

# Microphysical properties of refractory black carbon aerosols for different air masses at a central European background site

Yifan Yang[1], Thomas Müller[1], Laurent Poulain[1], Samira Atabakhsh[1], Bruna A. Holanda[2], Jens Voigtländer[1], Shubhi Arora[1], Mira L. Pöhlker[1,3]

[1]Leibniz Institute for Tropospheric Research, Leipzig, 04318, Germany
[2]Hessian Agency for Nature Conservation, Environment and Geology, 65203 Wiesbaden, Germany
[3] Faculty of Physics and Earth Sciences, Leipzig Institute for Meteorology, Leipzig University, 04103, Leipzig, Germany

Correspondence to: Yifan Yang (yang@tropos.de), Thomas Müller (muellert@tropos.de)

**Abstract:** Uncertainties remain in estimating black carbon's (BC) radiative forcing due to a limited understanding of its microphysical properties. This study investigated the physical properties of refractory black carbon (rBC) at the central European background site Melpitz during summer and winter, using a single particle soot photometer coupled with a thermodenuder. Different air masses associated with distinct rBC properties were identified in both seasons. In summer, rBC exhibited a similar mass concentration (~0.16 μg m$^{-3}$) among different air masses, with the smallest mass median diameter (MMD) of rBC overserved in the long transportation from the northwest (140nm), while in winter, the highest concentration (1.23 μg m$^{-3}$) and largest MMD (216 nm) were both observed in the air mass influenced by the easterly winds. Thickly coated rBC fractions increased during the daytime in summer, indicating the photochemical processes significantly influence the rBC mixing state. In winter, a higher fraction (27%) of rBC with thick coating in the cold air mass compared to the warm air mass (14%) suggests the contribution of residential heating emissions to the mixing state. Most rBC retained a low-volatile coating in the thermodenuder samples (58% mass fraction). In summer, photochemical processes also contribute to coating volatility, showing a higher fraction of rBC particles containing low-volatile coatings during the daytime. In winter, low-volatile coatings showed no significant diurnal variation and were more dependent on ambient temperature. Therefore, rBC coating volatility in winter is more influenced by emission sources, particularly residential heating, rather than atmospheric processes.

## 1 Introduction

Large amounts of black carbon (BC) are emitted into the atmosphere from the incomplete combustion of fossil fuels and biomass burning (Bond et al., 2013; Liu et al., 2020). BC  is the most absorbing atmospheric aerosol and therefore affects the Earth's climate system (Bond et al., 2006; Bond and Bergstrom, 2007). Compared with greenhouse gases such as carbon dioxide and methane, the atmospheric lifetime of BC is relatively shorter, in the range of hours to days (Bond et al., 2013). According to the sixth Intergovernmental Panel on Climate Change (IPCC, 2021) report, BC light absorption has significant regional effects. Additionally, BC can also be transported worldwide and contribute to the global climate





forcing as well (Hodnebrog et al., 2014). However, the estimation of BC direct radiative forcing ranged from +0.1 to +1.0 Wm$^{-2}$, leading to challenges in accurately assessing radiation absorption by aerosols in climate models (Wang et al., 2016). The uncertainties of BC climate forcing estimation can be attributed to a limited understanding of its size distribution, mixing state, morphology, spatiotemporal distribution, and absorption properties, all of which require more representative

and long-term measurements (Cappa et al., 2012; Bond et al., 2013; Liu et al., 2017; Romshoo et al., 2021). The size distribution of BC is mainly related to different sources, e.g. BC from biomass burning was found to have a larger peak volume equivalent diameter (VED) of approximately 200 nm, compared to urban emissions with a peak in VED of around 140 nm in Texas, US (Schwarz et al., 2008b). Zhang et al. (2020) also observed that the mass median diameter (MMD) of BC from diesel vehicles is smaller (~155 nm) than that from residential crop residue burning (~250 nm) and residential

firewood burning (~273 nm) in North China Plain. Freshly emitted BC is hydrophobic and independent from other atmospheric materials (Schwarz et al., 2008a). After emissions, BC is coated by other atmospheric components through condensation and coagulation (Liu et al., 2019; Sedlacek et al., 2022). Photochemical reactions were found to be important for the coating of BC, and the thickly coated BC particles tend to be internally mixed with secondary inorganic and organic components (Wang et al., 2017; Collier et al., 2018; Wang et al., 2019). Internally mixed BC becomes hygroscopic

and can act as cloud condensation nuclei (CCN) depending on its size and relative amount of water-soluble coatings (Rose et al., 2011). Laborde et al. (2013) observed that freshly emitted BC aerosols from traffic and biomass burning emissions were mostly non-hygroscopic and BC from aged air masses was more hygroscopic. Liu et al. (2013) observed pronounced hygroscopicity in BC-containing particles with ammonium nitrate in industrial areas of the UK.

Coating materials do not only affect the hygroscopicity of the BC particles. When non-absorbing materials coat on BC,

they can focus the incoming light onto the center of the particle, increasing BC core absorption efficiency. This coating-induced absorption enhancement, commonly referred to as the 'lensing effect', has been reported in many field measurements and laboratory studies, and the reported absorption enhancement due to the lensing effect can range from a factor of 1 (no lensing effect) to a factor of around 3 (Schwarz et al., 2008a; Cappa et al., 2012; Liu et al., 2015; Liu et al., 2017; Zhang et al., 2018). The variation of the lensing effect may be influenced by factors such as coating thickness

(Shiraiwa et al., 2010), coating composition (Saleh et al., 2014), and the BC core position within the particle (Wang et al., 2021b). The thermodenuder or catalytic stripper can heat the sample air from 100ºC to 400ºC, thus they are conventionally used in conjunction with instruments measuring particle light absorption to investigate the lensing effect (Nakayama et al., 2014; Ueda et al., 2016; Yuan et al., 2021). However, these studies usually assumed the coating of BC would be completely removed after passing through the thermodenuder or only considered the effect of non-refractory

coating on the absorption, but some low-volatile (LV) coating on BC can be found in the sample after being heated (Poulain et al., 2014), these low-volatile components would lead to the imprecise estimation of lensing effect (Shetty et al., 2021). On the one hand, the thermodenuder-based method would underestimate the lensing effect, because the absorption enhancement induced by these remaining LV coatings was considered as the absorption of the BC core. By





accounting for the LV coating, Zhang et al. (2023) found both in the field observation and model calculation, the real absorption enhancement can be as high as a factor of 2. On the other hand, if this LV-coating contains some absorbing components, e.g., brown carbon, in contrast, these coatings may not increase and even reduce the BC absorption at the ultraviolet spectral region (Feng et al., 2021; Luo et al., 2018). To facilitate the evaluation of climate change, additional research is required to gain a comprehensive understanding of the relationship between the mixing state and optical properties of BC.

Although numerous field studies exploring the mixing state of BC have been conducted in Europe, the majority of these researches have focused on short polluted periods or single seasons. To investigate the long-term variation of BC physical properties, continuous measurements were conducted at the central European background site Melpitz (Germany) since the end of July 2021. Mass concentration, size distribution, and mixing state of BC were measured by a Single Particle Soot Photometer (SP2, Droplet Measurement Technology, Longmont, CO, USA). A thermodenuder (300ºC) was

connected upstream of the SP2 to remove the volatility coating of BC. In this study, measurement results from August and December 2021, corresponding to the lowest and highest mass concentrations of BC observed during the measurement period, were selected to represent summer and winter, respectively. The present study aimed to analyze the differences in the physical properties of BC in relation to different air masses and atmospheric processes observed during distinct seasons.

**2 Method**

**2.1 Melpitz site**

Figure 1 shows measurements performed at the research site Melpitz (12°56'E, 51°32'N, 86 ma.s.l.) of Leibniz Institute for Tropospheric Research (TROPOS), 50 km to the northeast of Leipzig, Germany. The research site belongs to several observation networks such as GUAN (German Ultrafine Aerosol Network), ACTRIS (Aerosols, Clouds, and Trace gases

Research Infrastructure), EMEP (European Monitoring and Evaluation Programme), and GAW (Global Atmosphere Watch). The measurements at Melpitz are regarded as representative of the rural background conditions in Europe (Spindler et al., 2013; Atabakhsh et al., 2023). The meteorological parameters (wind speed, wind direction, temperature, and relative humidity) and aerosol parameters (e.g. mass and number concentrations, size distributions, optical properties, and chemical composition) were measured continuously. All online instruments are housed in the same laboratory

container and share a common aerosol inlet. This inlet line includes a $PM_{10}$ Anderson impactor positioned about 6 meters above ground level, followed by an automatic aerosol diffusion dryer, which actively maintains the relative humidity in the sampling line below 40% (Tuch et al., 2009). More detailed descriptions of the Melpitz site can be found, for example, in Spindler et al. (2012), Poulain et al. (2020), and Atabakhsh et al. (2023).



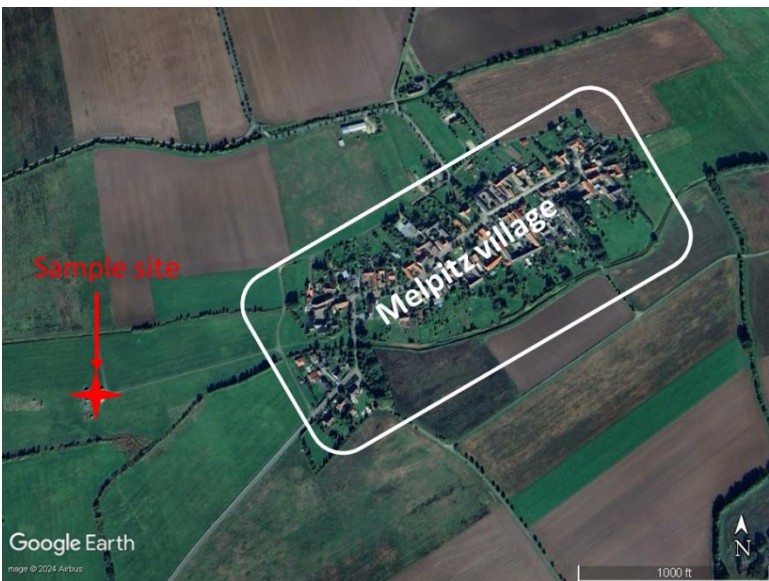


**Figure 1: Site plan of the Melpitz location. The map underneath is provided by ©Google Earth, 2024.**

## 2.2 Instruments

### 2.2.1 Single particle soot photometer (SP2)

The physical properties of individual BC particles were characterized by SP2. The principle of SP2 can be found

elsewhere (Schwarz et al., 2006; Mcmeeking et al., 2010). Here, briefly, SP2 is based on the laser-induced incandescence (LII) technique. In the SP2 measuring chamber, an intense continuous intracavity Nd: YAG laser beam is generated (at 1064 nm). When a sampled particle passes through the laser beam, its optical diameter can be obtained by the scattering light. If the particle contains absorbing compounds, it will absorb the laser radiation, be heated to its boiling point, and emit incandescent radiation. The refractory mass of individual particles is proportional to the incandescence signal. The

dominant absorbing component at this wavelength in the atmosphere is BC (Liu et al., 2014), and BC measured based on this LII technique is referred to as refractory black carbon (rBC). The mass concentration of rBC ($M_{rBC}$) is the sum of all single particle masses passing the chamber within a given time and flow rate. The mass equivalent diameter of rBC (as core diameter $D_c$) can be obtained by assuming the density of rBC of 1.8 g cm$^{-3}$ (Bond and Bergstrom, 2007). If a rBC particle with a small mass passes through the laser beam or the laser intensity is low, the conductive cooling to the

surrounding air dominates over the BC absorption. As a result, individual rBC particles cannot reach the temperature needed for emitting incandescent light. Besides, the incandescence signal of the large rBC particles can lead to the saturation of the detector. These detection limits of SP2 may cause an underestimation of rBC mass concentration (Schwarz et al., 2010). A commonly used method to eliminate the influence of the detection limit, the potential missing



mass fraction of rBC below and above the SP2 detection limit can be inferred from the extrapolation of a lognormal fit

of the measured rBC mass size distribution (Schwarz et al., 2006; Laborde et al., 2013; Pileci et al., 2021). In this study, the SP2 detection range for rBC cores is ~80-500nm, and the mean missing ratio of rBC mass concentration due to the SP2 detection limit in summer and winter were $17 \pm 7\%$ and $5\pm 4\%$, respectively.

A rBC-containing particle can be heated up to 4000k when it crosses the laser beam. The scattering signal of the rBC-containing particle will be distorted due to the evaporation of the volatile coating during the heating of the core. The

leading edge-only (LEO) fit method was used to rebuild the scattering signal, technical details about the LEO-fit approach can be found in Gao et al. (2007). The LEO fitted scattering signal and rBC core size were used to derive the optical diameter of BC-containing particle ($D_p$) by Mie calculations, with a core rBC refractive index =2.26–1.26i (Moteki et al., 2010) and a coating refractive index =1.5+0i (Liu et al., 2014). The mixing state of the rBC-containing particle can be quantified by the relative coating thickness $D_p/D_c$ or the absolute coating thickness (CT) $(D_p-D_c)/2$. The LEO-fit-derived

coating thickness can lead to nonphysical negative values. This phenomenon has been reported in other studies (Laborde et al., 2013; Taylor et al., 2015; Krasowsky et al., 2018; Ko et al., 2020), and can be attributed to many factors in addition to the noise of scattering and incandescence signal. For example, the extremely thin coating would emit a poor scattering signal, resulting in an underestimation of the rBC-containing particle size. The core size $D_c$ from the incandescent may be greater than the total particle size $D_p$. Moreover, if the rBC-containing particle is not a core-shell structure, rBC-free

materials only attached at the edge of rBC may also result in the 'negative coating thickness'. Taylor et al. (2015) also found that negative coatings can be caused by the used core density and refractive index. This study considers the rBC with negative CT as an 'uncoated' rBC. This definition will be discussed later. When considering the average coating thickness, these negative values were not counted. In addition to coating thickness, the mass fraction of coated rBC ($mf_{coated}$)was also used to quantify the mixing state of rBC:

$$mf_{coated} = \frac{M_{rBC-PCT}}{M_{rBC-CT}}, \tag{1}$$

The $M_{rBC-PCT}$ is the mass concentration of rBC containing the positive CT, while the $M_{rBC-CT}$ is the mass concentration of BC containing coating information, including both positive and negative CT.

Before field measurement, the scattering channel of SP2 was calibrated by spherical polystyrene latex (PSL) of known sizes, and the incandescence channel was calibrated using Aquadag®. Monodisperse Aquadag particles were selected by

a Centrifugal Particle Mass Analyzer (CPMA). A Differential Mobility Analyzer (DMA) was connected after the CMPA to remove multiple charged particles. Moreover, the detection efficiency of SP2 was derived by comparing it to a Condensation Particle Counter (CPC). For an Aquadag particle of ~0.34fg, corresponding to ~85nm mobility diameter, the SP2 detection efficiency was ~90 %. A further correction of 0.75 of the incandescent signal is required to represent the ambient particle suggested by Baumgardner et al. (2012)and Laborde et al. (2012).



### 2.2.2 Thermodenuder

The SP2 was connected to a thermodenuder (TD) system, allowing the measurement of the volatility of rBC-containing particles, as shown in Fig. 2. A detailed description of the TD design can be found in Wehner et al. (2002). A valve controlled the sample air passing through the TD or a bypass line (ambient line). The valve automatically switched every 10 minutes, alternatively delivering the TD sample or the ambient sample to SP2. The TD sample was heated to 300 °C. This temperature, on the one hand, is high enough to remove inorganic compounds (e.g. ammonium nitrate and ammonium sulfate) and most of the volatile organic components, on the other hand, this temperature is low enough to prevent the charring of organic compounds(Poulain et al., 2014). Thus, the majority of materials left in the TD sample were BC and a fraction of low-volatility oxygenated organic aerosol (LV-OOA) remained in the particle phase (Poulain et al., 2014).

rBC mass losses in the thermodenuder can be derived from the ratio of rBC mass concentrations from the ambient sample and TD sample, which accounts for ~20 % and ~11 % in summer and winter, respectively, as shown in Fig. s1 in Supplement. The denuder process did not obviously change the shape of rBC mass size distribution as shown in Fig. s2, only a slight tendency to shift of ~9nm towards smaller sizes was observed during the summer. The shape shifts of the rBC size distribution in TD samples did not influence our result significantly, thus we assume the size distributions for the ambient and TD samples are the same.

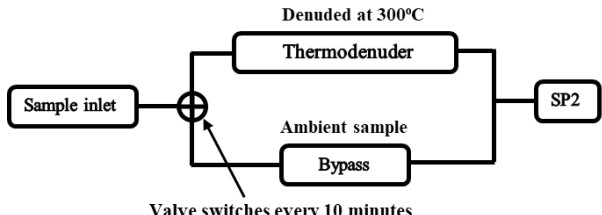

**Figure 2: Instruments setup of TD-SP2 system.**

### 2.2.3 ACSM and AMS

An aerosol chemical speciation monitor (ACSM, Aerodyne Research, MA, US; Ng et al., 2011) and an Aerodyne high-resolution time-of-flight aerosol mass spectrometer (HR-ToF-AMS, here referred to as AMS, DeCarlo et al., 2006 ) was used to measure the $PM_1$ bulk chemical composition of non-refractory aerosol species, including organics aerosols (OA), nitrate, sulfate, ammonium and chloride. The ACSM has been set at the Melpitz site since June 2012 and is measuring until nowadays with just a few data missing (Poulain et al., 2020; Atabakhsh et al., 2023). During summer, the chemical composition measured by AMS was used instead of ACSM, due to the higher time resolution obtained. More information about Melpitz ACSM and AMS can be found in Poulain et al. (2014), Poulain et al. (2020), and Atabakhsh et al. (2023).



## 3 Result

### 3.1 Overview of measurement

The time series of meteorological parameters, physical properties of rBC, and chemical compositions with a time
resolution of 1 hour during summer and winter were shown in Fig. 3. Aerosol components exhibited evident seasonal
variations. In summer the dominant composition was OA, and the mean mass fraction of OA was $55 \pm 13$ %. In contrast,
during winter, the OA fraction decreased to $29 \pm 14$ %, and the mass fraction of nitrate ($29 \pm 15$ %) significantly increased
compared with summer ($7.52 \pm 5.30$ %). Furthermore, the mass fraction of sulfate was $24 \pm 10$ % in summer and $13 \pm 8$ % in winter respectively.  As shown in Table 1, a much higher $M_{rBC}$ of $0.61 \pm 0.56$ µg m$^{-3}$ was observed in winter than
$0.16 \pm 0.12$ µg m$^{-3}$ in summer. The mean mass fraction of rBC is less than 5 % in both seasons. The MMD can be obtained
from the lognormal fit of the measured rBC mass size distribution, which can be considered to represent the overall size
of the rBC population for the given time window. The MMD in winter ($192 \pm 2$ nm) is significantly larger than in summer
($148 \pm 24$ nm) as shown in Fig. 3 (b) and Table.1, related to the different emission sources and air mass, which will be
discussed later.,

The average mass fraction of coated rBC ($mf_{coated}$) and coating thickness (CT) for rBC-containing particles in the SP2
detection range is exhibited in Fig. 3 (c) and (d).  In the ambient sample, CT did not show an evident difference between
the two seasons, which is contrary to previous research usually observed more thickly coated rBC in winter (Yang et al.,
2019; Liu et al., 2019; Kompalli et al., 2020). TD can remove most of the volatile coating of rBC, but the remaining
coatings were still found on rBC particles in the TD sample. On average, 58 % of rBC masses are still coated after being
heated up to 300 ℃ in both seasons, these remaining coatings are referred to as low-volatile coatings (LV-coating). The
variation of CT and $mf_{coated}$ can be explained by the size range selected to quantify the mixing state and the atmospheric
processes (e.g., photochemical process, hygroscopic growth), a detailed discussion will be presented in the next chapter.



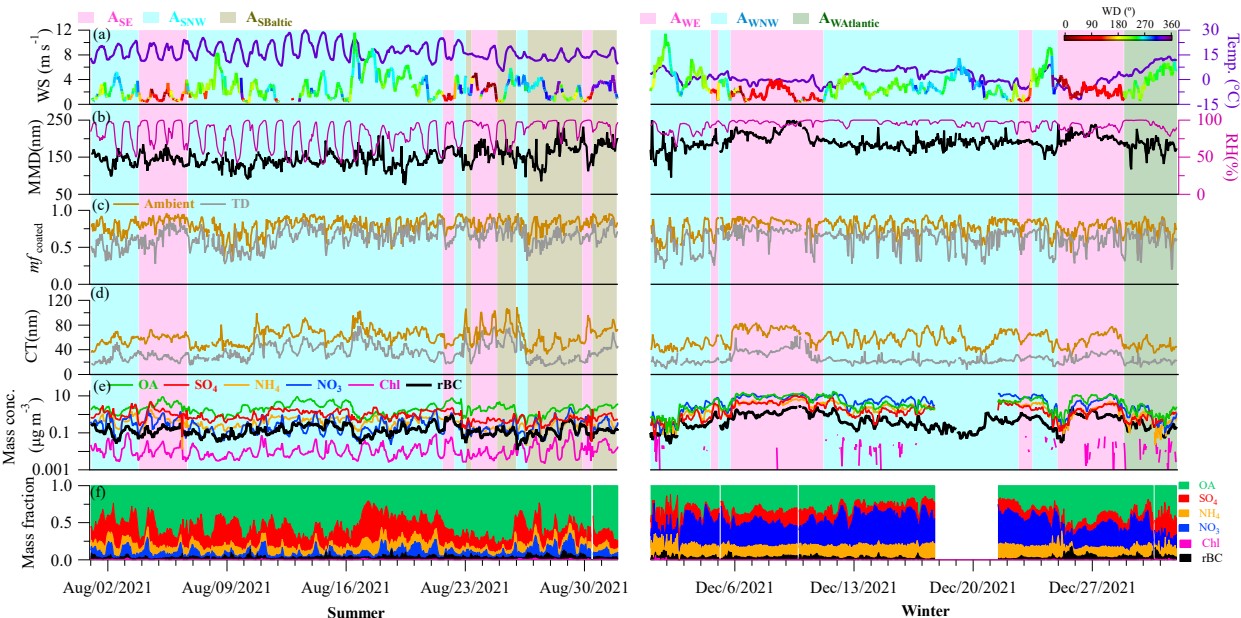

**Figure 3: Time series of all measurements in summer and winter: (a) wind speed (WS), wind direction (WD) and temperature; (b) mass median diameter (MMD) of rBC size distribution and relative humidity (RH); (c) mean mass fraction ($mf_{coated}$) and (d) coating thickness (CT) of coated rBC in ambient and TD sample; (e) mass concentration and (f) mass fractions of aerosol-phase chemical components and rBC. The shaded areas indicate air masses.**

**Table 1. Seasonal mean values and variability of meteorological parameters and physical properties of rBC in Fig. 3**

| | | WS (m s$^{-1}$) | Temp. (ºC) | RH (%) | $M_{rBC}$ (µg m$^{-3}$) | MMD (nm) | CT (nm) Ambient | CT (nm) TD | $mf_{coated}$ Ambient | $mf_{coated}$ TD |
|---|---|---|---|---|---|---|---|---|---|---|
| **Summer** | **Range** | 0.22~11.60 | 4.9~29.9 | 40.5~100 | 0.01~0.89 | 75~244 | 32.23~108.83 | 12.94~72.12 | 0.31~0.97 | 0.28~0.90 |
| | **Mean** | 2.55 ± 1.79 | 16.73 ± 4.71 | 82.16 ± 17.00 | 0.16 ± 0.12 | 148 ± 24 | 62.01 ± 13.80 | 35.80 ± 13.68 | 0.80 ± 0.11 | 0.63 ± 0.13 |
| **Winter** | **Range** | 0.37~11.40 | -12.0~13.9 | 64.0~+100 | 0.03~2.93 | 95~283 | 25.70~83.70 | 7.98~63.38 | 0.39~0.94 | 0.12~0.87 |
| | **Mean** | 2.98 ± 1.84 | 1.68 ± 4.76 | 94.17 ± 6.00 | 0.61 ± 0.56 | 192 ± 21 | 57.39 ± 12.39 | 25.32~8. 28 | 0.78 ± 0.11 | 0.52 ± 0.16 |


Figure. 4 shows clusters of 72-hour air mass back trajectories at an altitude of 100 meters above the ground level, along with the non-parametric wind regression (NWR) analysis of rBC properties following the procedures described in Petit et al. (2017). The wind speed (WS) and wind direction (WD) were measured above 6 meters above the ground. Three different air masses, associated with distinct rBC properties, were categorized based on the trajectory clusters and wind

analysis for both seasons, as summarized in Table 4. As shown in Fig.4 (a) and (b), three different clusters of back trajectories were classified. During summer, two trajectory clusters from the northwest direction can be found, Cluster1 of summer (Cs1) passing the North Sea, Netherlands, and Northwest of Germany, and Cluster2 of summer (Cs2) passing northern France and western Germany. The rBC properties of Cs1 showed no significant differences from those of Cs2. Consequently, $A_{SNW}$ was used to represent the air masses including these two clusters arriving from the northwest in



summer. Moreover, $A_{SBaltic}$ is indicated by Cluster3 of summer (Cs3) originating from Finland and passing the Baltic Sea in the southwest direction. In winter, Cluster1 and Cluster2 of winter (Cw1 and Cw2) followed paths similar to Cs1 and Cs2, thus being identified as air masses from the northwest in winter ($A_{WNW}$). Furthermore, Cluster3 of winter (Cw3) was classified as $A_{WAtlantic}$, crossing northern France from the Atlantic Sea, and then following the same path as Cw2 in Germany.

All trajectory clusters arrived in the northwestern or southwestern sections of Melpitz. In addition, as shown in Fig.4 (c) and (d), the dominating WD sector for the Melpitz site is southwest, with a WS higher than 5 m s$^{-1}$ in both seasons, as shown in the joint probability distribution of WS and WD. However, the high $M_{rBC}$ can be observed in the east section of NWR for both seasons. During summer, a high concentration of rBC can be observed in the northeast particularly when wind speeds are below 5 m s$^{-1}$, indicating local emissions from the village of Melpitz or the city of Torgau (5km), and

these rBC particles exhibited smaller size (small MMD) and thin coating (small CT). Additionally, another high concentration of rBC from the southwest may relate to the emission from the village of Klitzschen (3km). The presence of a thick coating and large MMD of rBC observed from the northwest section accompanied by high wind speeds may relate to $A_{SBaltic}$. In winter, high rBC concentration was principally observed from the northeast but also with a significant contribution from the southeast, occurring under both low (<5 m s$^{-1}$) and high (>5 m s$^{-1}$) wind speeds, suggesting both

local emission and transportation from the east. Concurrently, rBC of larger size (large MMD) and with thicker coatings (high CT) was also observed from similar directions. Therefore, the eastern section shown in the NWR analysis was defined as the air mass related to the easterly winds, with $A_{SE}$ representing this in summer and $A_{SW}$ in winter, respectively.





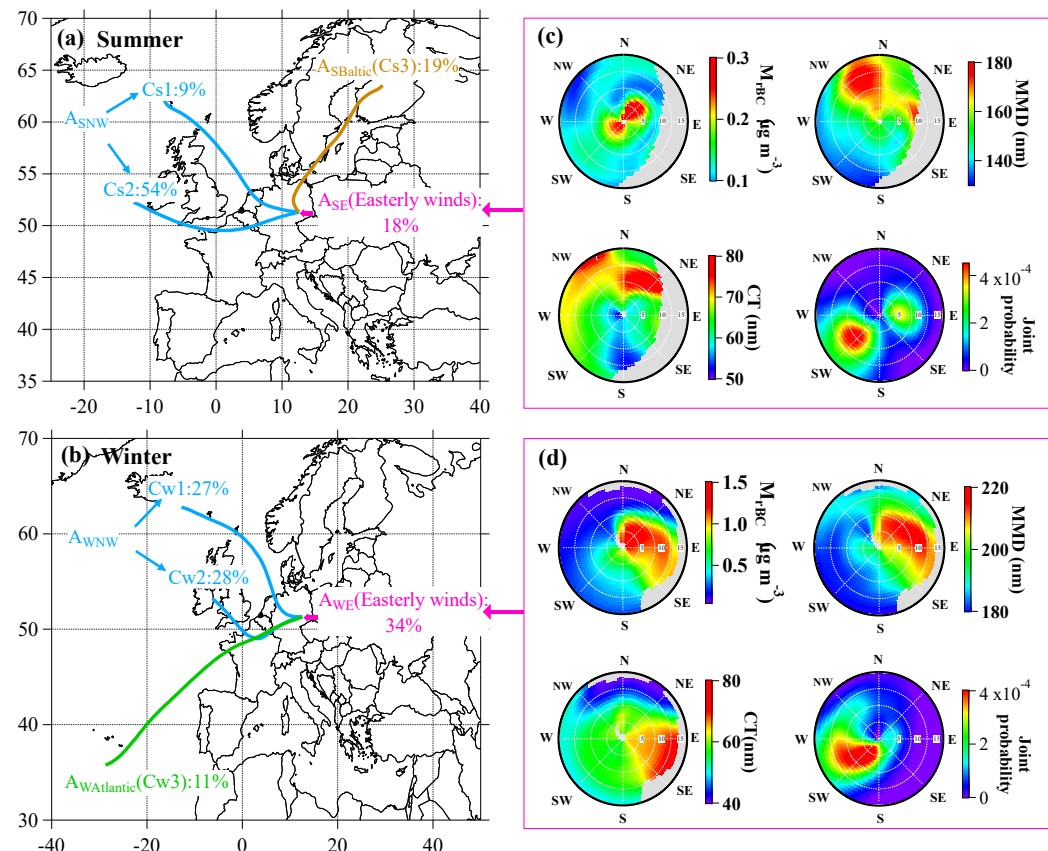

Figure 4: The air masses identified from clustered backward trajectories (a)&(b) and the wind analysis on mass concentrations, MMD and CT of rBC, and joint probability distribution of wind speed and wind direction (c)&(d) in summer and winter. The radius and angle of wind analysis refer to wind speed (m s$^{-1}$) and wind direction.

Table 2: Classification of air masses based on the clustered backward trajectories and wind analysis

| | Summer | | Winter |
|---|---|---|---|
| $A_{SNW}$ | Air mass of long transportation from the northwest direction (Cs1 & Cs2 of Fig. 4(a)) | $A_{WNW}$ | Air mass of long transportation from the northwest direction (Cw1 & Cw2 of Fig. 4(b)) |
| $A_{SBaltic}$ | Air mass of long transportation through the Baltic Sea (Cs3 of Fig. 4(a)) | $A_{WAtlantic}$ | Air mass of long transportation through the Atlantic Sea (Cs3 of Fig. 4(b)) |
| $A_{SE}$ | Air mass influenced by the easterly winds (East section of Fig. 4 (c)) | $A_{WE}$ | Air mass influenced by the easterly winds (East section of Fig.4 (d)) |




### 3.2 Mass concentration and size distribution of rBC

Figure 5 (a) and (b) exhibit the $M_{rBC}$ for different summer and winter air masses respectively. In summer, relatively similar $M_{rBC}$ levels were found, with $A_{SE}$ (mean $M_{rBC}$ of 0.18 ± 0.17 µg m$^{-3}$) being only slightly higher than $A_{SNW}$ and $A_{SBaltic}$ (both with mean $M_{rBC}$ around 0.16 µg m$^{-3}$). In winter, rBC exhibited distinct different $M_{rBC}$ among $A_{WE}$ (1.23 ± 0.60 µg m$^{-3}$), $A_{WNW}$ (0.36 ± 0.32 µg m$^{-3}$), and $A_{WAtlantic}$ (0.23 ± 0.12 µg m$^{-3}$). For all air masses during summer, diurnal cycles of $M_{rBC}$ exhibited low concentrations during the daytime and higher concentrations at night. This diurnal cycle is mainly related to planetary boundary layer development. Besides, the traffic emissions during rush hours from the Bundesstrasse B87 located approximately 1 km north of the station, may also contribute to the peak concentration. In winter, $A_{WE}$ displayed the most evident concentration peak around 20:00 while the peak of the other air masses was observed around 18:00. Similar diurnal variation in both seasons has been reported in other studies in Germany, e.g. Sun et al. (2019) and Atabakhsh et al. (2023).

The rBC mass size distributions for summer, shown in Fig. 6, with the MMD of $A_{SE}$ (164nm) and $A_{SNW}$ (140nm) are consistent with the rBC size ranges from 135 nm to 167 nm derived from mixing sources of solid fuel (wood and coal burning) and liquid fuel (traffic emission) by Liu et al. (2014). According to van Pinxteren et al. (2024), 40 % of the continuously running central heating systems in Melpitz village are fueled by solid fuel (wood and coal) and 46 % by liquid fuel(oil and liquid petroleum gas) which was considered as representative of the domestic heating system for the region. Furthermore, Atabakhsh et al. (2023) found biomass burning organic aerosol (BBOA) to have a relatively good correlation with BC. BBOA in summer was linked to water heating systems using wood briquettes and logs, recreational open fires, or barbecue activities at Melpitz (van Pinxteren et al., 2020). The traffic emission from nearby roads and the city of Torgau (with approx. 20000 inhabitants, the north-east direction in a 7 km distance) could be one of the potential emission sources of rBC as well. In addition, as shown in Fig. s3, a higher fraction of $A_{SNW}$ trajectory traces originated from high altitudes and arrived at Melpitz at lower latitudes. This suggests that the smaller MMD of $A_{SNW}$ may be related to relevant cloud processing during transport could remove particles containing larger rBC cores (Moteki et al., 2012; Che et al., 2022). The largest MMD (176 nm) in summer was observed during $A_{SBaltic}$, which may be associated with most of the trajectory traces traveling at low altitudes (Fig. s3). Certain air masses originating from the Baltic region and passing through Poland may transport aerosols from coal combustion (Atabakhsh et al., 2023). These emission sources are known to contain rBC of large sizes(Liu et al., 2014; Zhang et al., 2020).

In winter, $A_{WE}$ demonstrated a distinctly larger MMD (216 nm) compared to $A_{WNW}$ (187 nm) and $A_{WAtlantic}$ (185 nm), respectively. Similar MMDs of rBC were observed in Brazil biomass burning plumes ranging from 180nm to 226nm (Holanda et al., 2023). Zhang et al. (2020) showed that rBC from residential firewood (MMD of 273nm) is much larger than diesel vehicle emissions(MMD of 155nm). Liu et al. (2014) also observed that rBC from residential solid fuel burning for space heating purposes (such as coal or wood burning) is larger than traffic-emitting rBC. Therefore, residential heating could be an important emission source of rBC at Melpitz in winter. Thus, the lowest temperature (-2.14 ± 3.34



ºC) during $A_{WE}$ (Fig. s4) could contribute to the largest MMD, as the decrease in ambient temperature would increase the

proportion of rBC emitted from residential heating sources, as well as an increase in the size of the rBC. The relatively

high wind speed with a high concentration of large rBC particles can be found in Fig. 4, which suggests transportation

from the Czech Republic and Poland. This observation corresponds with the findings of Atabakhsh et al. (2023), who

found that high concentrations of coal combustion organic aerosol(CCOA) and BBOA were transported from the eastern

direction during winter which were strongly correlated with BC.


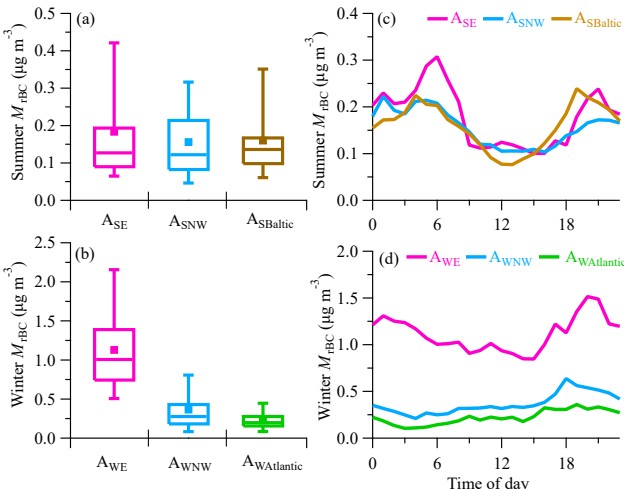

**Figure 5: Statistical analysis and diurnal variation of rBC mass concentration for each air mass in winter and summer. The upper and lower edges of the box denote the 25 % and 75 % percentiles, respectively. The middle line and square markers indicate the median and average values with error bars indicating the 10 % and 90 % percentiles.**






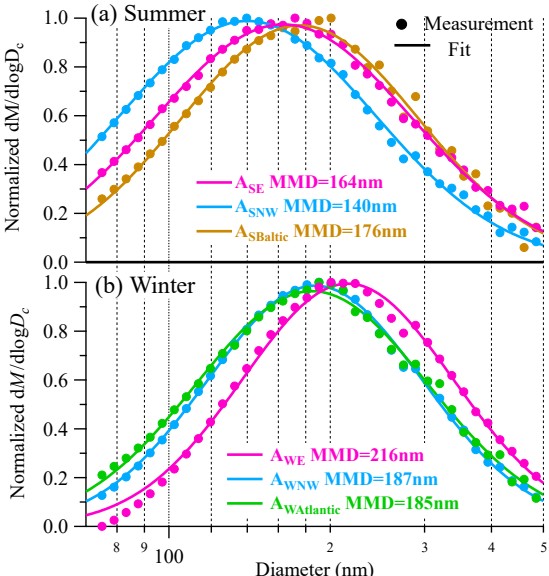

**Figure 6: Mass size distribution of rBC for each air mass in (a) summer and (b)winter. The circles show measurements, and the solid line represents the lognormal fit.**

### 3.3 Mixing state of rBC-containing particles

Most rBC particles larger than 80 nm observed in this study were coated, with significantly varying CT for different conditions. However, as noted in Section 2.2.1, negative CT was observed for some particles. In the ambient sample, the mass fraction of rBC with negative coating only accounts for 14% of all measured rBC particles in summer and 19 % in winter. This ratio increased up to 24 % and 30 % for the TD sample. The increasing ratio of rBC with 'negative coating' is associated with the removal of volatile coating in the denuder process. Thus, we assume that all the rBC particles with negative coating in this study are the 'uncoated rBC', and the 'coated rBC' indicated the rBC containing positive LEO-fit derived coating thickness. The mixing state discussed in this chapter indicated the $mf_\text{coated}$ and CT of rBC in the ambient sample.

#### 3.3.1 Size-resolved coating thickness

Figure 7 shows the size-resolved CT of the rBC during summer and winter. The color indicates the total volume of rBC-containing particles of each grid point with:

$$\text{Volume}_{grid(i,j)} = \frac{\pi}{6} D_{p(i,j)}^3 N_{i,j} = \frac{\pi}{6} (D_{c,i} + 2 \times CT_j)^3 N_{i,j}, \tag{2}$$



wherein $D_p$ is the diameter of the rBC-containing particle. $N$ is the total number of rBC particles, and $i$ and $j$ represent the $i$-*th* bin of core diameter (x-axis) and *j-th* bin of CT (y-axis) respectively.

Different types of rBC were categorized based on their core size and CT. The area labeled 'Small rBC without coating information' (rBC$_{small}$), as shown in the bottom-left slashed region of each panel of Fig. 7, indicates rBC particles that exhibited neither a positive CT nor a negative CT. The particles with negative coating thickness were defined as 'Uncoated rBC (rBC$_{uncoated}$)'. Moreover, in both seasons, two distinct regions associated with a high volume of coated rBC (highlighted in red) are evident in all air masses except A$_{WAtlantic}$. The bottom-right region consists of rBC with a relatively

thin coating thickness, where the relative coating thickness ($D_p/D_c$) is smaller than 1.5, defined as 'Thinly coated rBC (rBC$_{thin}$)'. In the upper red region, where the rBC particles contain thicker coatings, 'Moderately coated rBC (rBC$_{moderate}$)' refers to the particles with CT less than 150 nm, while the 'Thickly coated rBC (rBC$_{thick}$)' donates those with CT greater than 150 nm. For rBC cores larger than 300 nm, most particles exhibited negative (rBC$_{small}$) or thin coatings (rBC$_{thin}$), though some were observed with relatively thick coatings. These thickly coated particles were classified as extremely

large coated rBC (rBC$_{ex-large}$).

The rBC$_{small}$ is mainly due to the detection limit of the SP2 scattering detector. The detector can only discern the scattering signal from rBC-containing particles when the diameter of the rBC core is considerably large or when the coating on the rBC is thick enough. This detection limitation hinders the ability to derive coating information from scattering light for smaller rBC particles. Consequently, the mixing state of a large fraction of small rBC remains unaccounted for. As shown

in Fig. 8, the core mass fraction of rBC$_{small}$ during summer (25 %~33 %) is higher compared to winter (14 %~24 %), while rBC$_{uncoated}$ showed contrasting seasonal variation to rBC$_{small}$ (14 %~15 % in summer and 18~23 % in winter). This observation is associated with the smaller size distribution of the rBC core during summer. The missing CT information of rBC$_{small}$ would bias the average CT calculation. In the case of coated rBC smaller than 150 nm, only the relatively thick coating was counted, therefore average CT would be overestimated due to the missing of the thin coating values.

Ko et al. (2020) also discussed the bias of average CT induced by the scattering detection.

The rBC$_{thin}$ and rBC$_{uncoated}$ may be associated with the liquid fuel combustion according to similar patterns of size-resolved coating thickness of rBC quantified by different quantities (scattering enhancement, particle numbers) in some other studies (Liu et al., 2014; Liu et al., 2019; Brooks et al., 2019; Zhang et al., 2020). The liquid fuel combustion is associated with the central heating system of Melpitz and traffic emissions. As shown in the Fig. 8, rBC$_{thin}$ accounts for the highest

mass fraction for all air masses in both seasons, which implies that liquid fuel combustion is one of the primary emission sources of rBC at Melpitz.

The rBC$_{moderate}$, rBC$_{thick,}$ and rBC$_{ex-large}$ can be linked to the aged rBC or other emission sources such as biomass burning and coal combustion (Liu et al., 2014; Zhang et al., 2020). The core mass fractions of rBC$_{moderate}$ during summer (13 %~14 %) were slightly lower than A$_{WE}$ (18 %) and A$_{WNW}$ (16%) during winter. However, the core mass fraction of

rBC$_{moderate}$ during summer correlated to the small size distribution of rBC, as some of the coated rBC is under the detection




limit of SP2, resulting in the higher core mass fraction of $rBC_{small}$ and the reduction in the core mass fraction of $rBC_{moderate}$. As for the $rBC_{thick}$ region, a higher volume of rBC with thicker CT can be observed in $A_{SNW}$ and $A_{SBaltic}$ compared to $A_{SE}$. During winter, the high volume of $rBC_{thick}$ is exclusively observed in $A_{WE}$, with the highest core mass fraction of $rBC_{thick}$ (5%) and $rBC_{ex-large}$ (4%). These observations correlate with the lowest temperature of -2.14 ± 3.34ºC (Fig. s4) and

possibly increased residential heating emissions. In contrast, only the high volume of $rBC_{thin}$ was observed in $A_{WAtlantic}$, associated with the highest temperature of 8.50 ± 4.00ºC and less heating emissions.

The size-resolved CT revealed the bias in average CT caused by the detection limit of SP2, offering a more direct observation of the mixing state of rBC compared to average CT. For example, in summer, a high volume of $rBC_{moderate}$ was observed only at smaller rBC core sizes (below approximately 150 nm). For rBC core sizes larger than 150 nm, thick

coatings were less observed, particularly in $A_{SE}$. In winter, however, a high volume rBC containing thick coatings (greater than 150 nm) was observed for core sizes between approximately 120 nm and 220 nm in $A_{WE}$. This distribution of non-BC material across the BC-containing particle population of different core sizes could be referred to as mixing state heterogeneity, as reported in some studies (Zhao et al., 2019; Zhai et al., 2022). Such heterogeneity plays an important role in investigating the absorption enhancement of coated rBC (Fierce et al., 2016; Fierce et al., 2020; Zeng et al., 2024).

Moreover, size-resolved CT could also provide insights into the source apportionment of rBC, which requires further investigation.

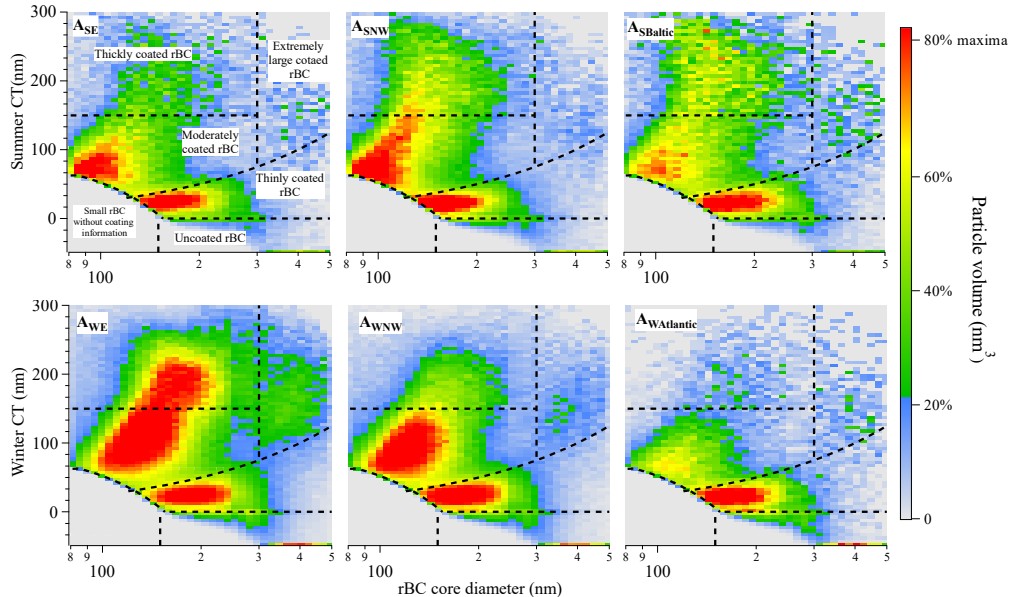

**Figure 7: Coating thickness as a function of rBC core size for each air mass of the ambient sample in summer and winter. Each plot is colored by the rBC-containing particle volume (note that the color scale changes between the subplots. The color scale**

**was set to be red when the volume is above 80 % of the maxima for each air mass). The black dash lines from the top to the bottom are y=300, x=150, y=0.25x ($D_p/D_c$=1.5), y=140.5-0.89x, and y=150.**



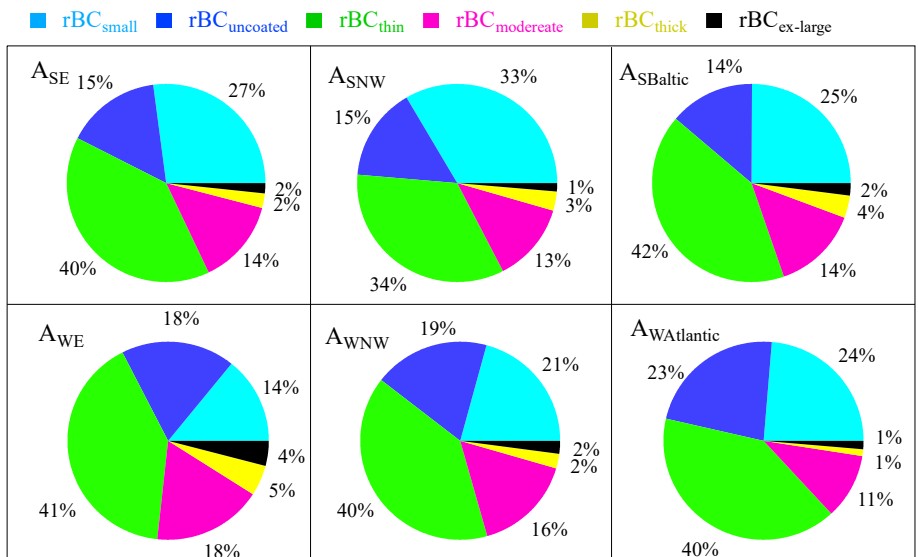

**Figure 8: The rBC core mass fraction of different types of rBC of different air masses in Fig. 7.**

### 3.3.2 Diurnal variation of size-resolved coating thickness

Figure 9 displays the diurnal variation of size-resolved coating thickness in different air masses. During summer, a high volume (red) of rBC$_{thin}$ can be observed most of the day. This observation is associated with the liquid fuel combustion of the continuously running heating system of Melpitz village (van Pinxteren et al., 2024). However, during evening rush hours after 18:00, the increasing volume in rBC$_{thin}$ and rBC$_{uncoated}$ regions can be found in the size-resolved CT plot for A$_{SE}$ and A$_{SNW}$. As for A$_{SBalitc}$, this increase can be observed after 15:00, which is inconsistent with the mass concentration peak of rBC, shown in Fig. 5 (c). Simultaneously, the total mass fraction of rBC$_{thin}$ and rBC$_{uncoated}$ during these times increases by approximately 10 % for all air masses compared to noon. Therefore, the mixing state of rBC could be also influenced by the traffic emissions during rush hours.

Throughout the daytime for all air masses during summer, a continuous increase in particle volume in the rBC$_{moderate}$ and rBC$_{thick}$ region is observed, accompanied by a decrease in the rBC$_{thin}$ region. Additionally, the rBC$_{moderate}$ and rBC$_{thick}$ exhibit higher rBC core mass fractions between 9:00 and 17:00, with their total mass fraction reaching a maximum of ~21 % for A$_{SE}$ and A$_{SNW}$ and ~25 % for A$_{SBaltic}$. This observed transition to the thicker coating region can be regarded as evidence of coating growth of rBC-containing particles, implying a significant role of photochemical reactions in the aging process of rBC, which has been reported in other studies(Liu et al., 2014; Yang et al., 2019). Moreover, the red area was observed to cross from the rBC$_{moderate}$ region to the rBC$_{thick}$ region as early as 09:00 for A$_{SNW}$ and 06:00 for A$_{SBaltic}$. However, for A$_{SE}$, this transition can be only found in the afternoon. This could be attributed to the more local emissions of freshly emitted rBC particles in A$_{SE}$.



During winter, diurnal variation of $rBC_{thin}$ and $rBC_{unocated}$ is similar to the summertime for all air masses. In contrast, high volumes of $rBC_{moderate}$ or $rBC_{thick}$ were observed at night for $A_{WE}$ and $A_{WNW}$, which may relate to the cold temperature

with more residential heating emissions. Temperature can be considered as a crucial factor influencing the mixing state of rBC in winter at Melpitz, giving the impact on the residential heating emissions. The high-volume region of the $rBC_{thick}$ was exclusively observed in $A_{WE}$, correlating with the coldest temperature. While during the warmest $A_{WAtlantic}$, even the high volume of $rBC_{moderate}$ was absent. In addition, the mass fraction of $rBC_{moderate}$ and $rBC_{thick}$ exhibited a consistent decreasing trend between 6:00 and 20:00 in $A_{WE}$. However, this decline during the daytime was interrupted at noon in

$A_{WNW}$. The total mass fraction of $rBC_{moderate}$ and $rBC_{thick}$ peaked at 24 % between 3:00 and 5:00, then consistently decreased to 18 % by 11:00. At noon, this mass fraction of more coated particles climbed back to 20 %, followed by a continuous decline to 14 % by 17:00. Similar variation was also observed in $A_{WAtlantic}$, which implied the photochemical process could also affect the mixing state of rBC-containing particles from west long transportation in winter. However, this impact is not as significant as in summer.






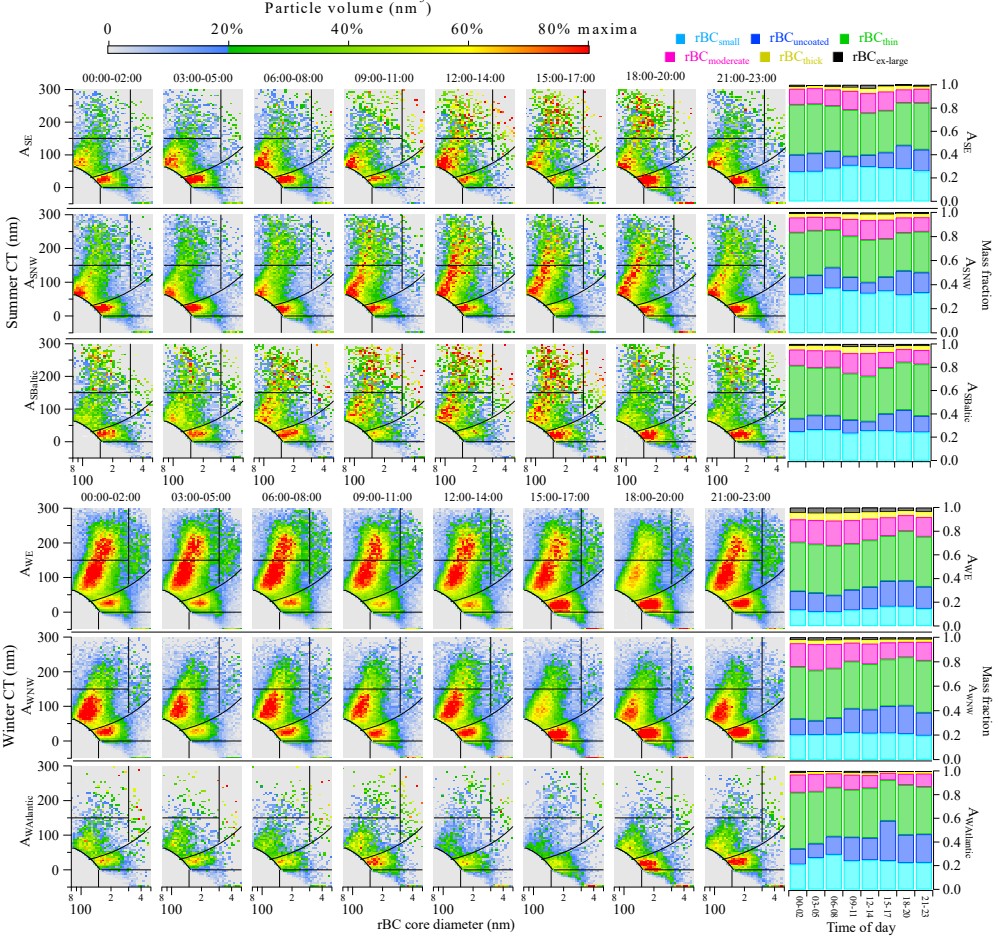

**Figure 9: Diurnal variation of size-resolved coating thickness and mass fraction of each type of rBC in the ambient sample. The panels for each air mass, from left to right, represent time intervals of 00:00–02:00, 03:00–05:00, 06:00–08:00, 09:00–11:00, 12:00–14:00, 15:00–17:00, 18:00–20:00, and 21:00–23:00. The color scale represents the total particle volume, with red set at**
**80% of the maximum value. In some panels, the red color scale was adjusted below 80% of the maximum to minimize the impact of exceptionally large values.**

To better quantify the mixing state of bulk rBC populations, analysis of average CT and $mf_{coated}$ (1-hour time resolution) with the core size range selected between 80~500nm were shown in Fig. 10 and Table.3. In summer, mean CT did not
exhibit significant differences among different air masses. Mean CT about 62 nm for all air masses, can be affected by the detection limit of SP2 and the smaller size of the rBC core during summer. However, the 75th percentile CT in $A_{SE}$ is lower than the other two air masses during summer. This result is consistent with the finding that a lesser fraction of thickly coated rBC was observed during $A_{SE}$. In winter, the influence of the detection limit seems to be less significant.





The average CT agreed with the observation of the size-resolved CT, the highest average CT of $63.16 \pm 12.65$nm was
observed during $A_{WE}$, whereas the smallest value of $45.98 \pm 6.13$nm was found during $A_{WAtlantic}$. As shown in Fig. 10 (b),
the peak of CT was found during the daytime in summer, whereas it occurred at night in winter. Specifically, in summer,
the CT peak of $A_{SE}$ was observed in the afternoon, while the peak of $A_{SBaltic}$ was found in the morning. As for $A_{SNW}$, a
relatively stable high value was observed throughout all daytime. In contrast to summer, CT exhibited similar diurnal
cycles among different air masses in winter. Therefore, average CT can effectively capture the varying tendency of the
mixing state of rBC displayed in Fig. 9, although the absolute value was influenced by the absence of coating information
for rBC below the detection limit.

In addition to CT, $mf_{coated}$ was used to quantify the mixing state of rBC as well. In general, as shown in Fig. 10 (a) and
Table.3, most of the rBC-containing particles can derive positive CT in both seasons. In summer, the average $mf_{coated}$ of
$A_{SNW}$ ($0.78 \pm 0.12$) is lower than that of the other two air masses ($0.83 \pm 0.07$ for $A_{SNW}$ and $0.85 \pm 0.06$ for $A_{SBaltic}$).
Similar to CT, diurnal variation of $mf_{coated}$ exhibited high value during daytime for $A_{SE}$ and $A_{SNW}$. However, a 'delay' of
the peak can be observed between the diurnal variation of CT and $mf_{coated}$. Particularly in $A_{SE}$, the peak of $mf_{coated}$ was
observed in the morning while the peak of CT was found in the afternoon. This non-simultaneous variation between CT
and $mf_{coated}$ may relate to local emission and atmospheric processes, which require further investigation.

The diurnal variations of sulfate, nitrate, and OA mass concentrations are shown in Fig. 10 (c). In summer, sulfate
exhibited similar variations to $mf_{coated}$. The sulfate mainly resulted from the condensation of $H_2SO_4$ produced by the gas
phase photooxidation of $SO_2$(Poulain et al., 2011; Zhang et al., 2015). On the one hand, the increasing $H_2SO_4$ could serve
as an indicator of the photochemical processes in the atmosphere. On the other hand, in addition to organic materials,
sulfate may have an impact on the coating of rBC at Melpitz. The contribution of sulfate to rBC mixing state can also be
found in other studies(Xu et al., 2018; Wang et al., 2021a). In addition, during winter, an increasing concentration of
sulfate can be also observed during daytime in $A_{WNW}$ and $A_{WAtlantic}$, giving evidence of the aforementioned photochemical
process in winter. This increasing mass concentration of sulfate is consistent with the discussion on the increase in the
mass fraction of $rBC_{moderate}$ and $rBC_{thick}$ in section 4.2.2. Furthermore, the impact of heterogeneous reactions on the mixing
state of rBC should be noticed in winter, such as the formation of nitrate and OA at night (Pathak et al., 2009; Zhao et al.,
2017). Besides, hygroscopic growth may also contribute to the coating process of rBC, as higher RH and aerosol
hygroscopicity were observed at Melpitz in winter (Wang et al., 2022).





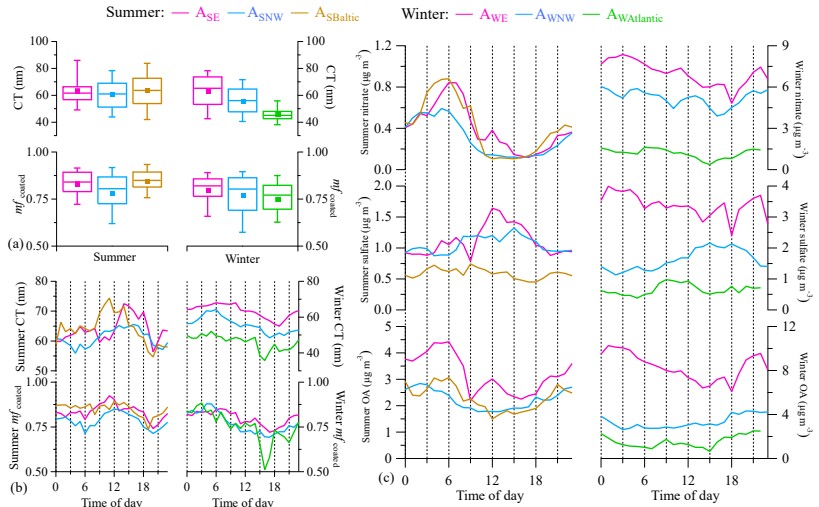

**Figure 10: (a) Statistical analysis and (b) diurnal variation of average CT and $mf_{coated}$ of rBC in the ambient sample. (c) Diurnal variation of chemical composition mass concentration.**

**Table 3: 25th percentile, 75th percentile, and mean values of $mf_{coated}$ and CT in Fig. 9(a).**

| | | Summer | | | Winter | | |
|---|---|---|---|---|---|---|---|
| | | $A_{SE}$ | $A_{SNW}$ | $A_{SBaltic}$ | $A_{WE}$ | $A_{WNW}$ | $A_{WAtlantic}$ |
| **CT (nm)** | $25^{th}{\sim}75^{th}$ | 56.17~66.84 | 50.68~69.45 | 52.23~73.24 | 52.71~74.23 | 47.19~65.21 | 41.77~64.92 |
| | **Mean** | 63.59 ± 12.92 | 61.04 ± 13.38 | 63.89 ± 15.79 | 63.16 ±12.65 | 57.93 ±10.24 | 45.98 ± 6.13 |
| $mf_{coated}$ | $25^{th}{\sim}75^{th}$ | 0.79~0.90 | 0.72~0.87 | 0.81~0.90 | 0.76~0.86 | 0.69~0.87 | 0.69~0.83 |
| | **Mean** | 0.83 ± 0.07 | 0.78 ± 0.12 | 0.85 ± 0.06 | 0.80 ± 0.09 | 0.77 ± 0.12 | 0.75 ± 0.10 |

.

### 3.4 The coating volatility of rBC-containing particles

Analyzing the mixing state on the TD sample was performed to investigate the coating volatility of rBC. As shown in Fig. 11, the thickly coated rBC was rarely observed after passing through the thermodenuder, and the core mass fraction of

$rBC_{thick}$ is less or around 1 % for all air masses, as indicated in Fig. 12. This observation implied the dominant components of thick coating are volatile. Due to the loss of volatile coating during the denuder process, some particles shrunk in size, falling under the detection limit, while some other particles either lacked or contained minimal LV-coating. Consequently, compared to the ambient sample, the core mass fraction of $rBC_{small}$ and $rBC_{uncoated}$ increased in the TD sample. The majority of rBC-containing particles observed in Fig. 11 were $rBC_{thin}$, although the reduced core mass fraction of $rBC_{thin}$

in the TD sample ranged from 8 % to 13 % compared to the ambient sample. During summer, the lowest fraction $rBC_{thin}$ (29 %) was observed in $A_{SNW}$, related to the smaller size distribution. However, a relatively high volume (green color) of





rBC$_{moderate}$ can be observed in A$_{SNW}$. During winter, A$_{WE}$ exhibited distinct size-resolved CT differing from A$_{WE}$ and A$_{WAtlantic}$, with a relatively high volume observed in the rBC$_{moderate}$ region, with the highest core mass fraction of 8%.

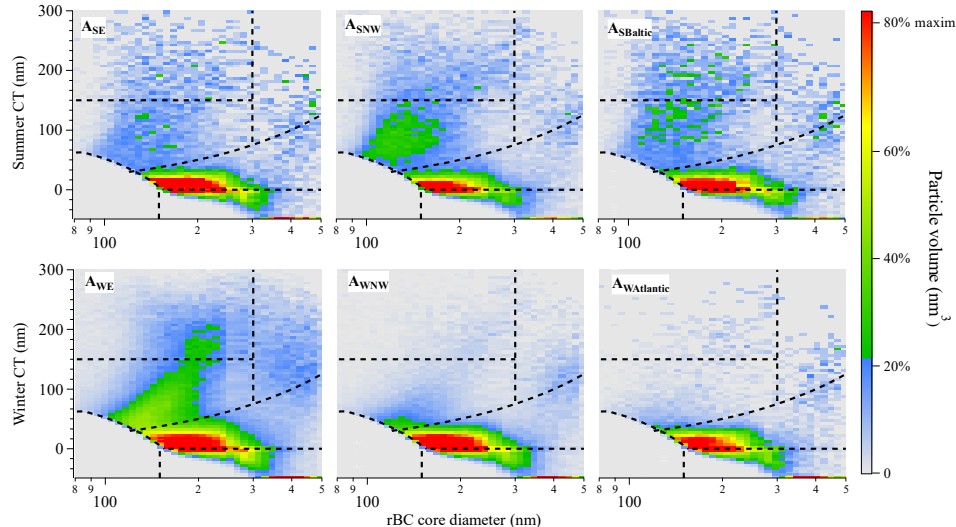

**Figure 11: Coating thickness as a function of rBC core size for each air mass of the TD sample in summer and winter. The color bars and black dash lines are the same as Fig. 7**

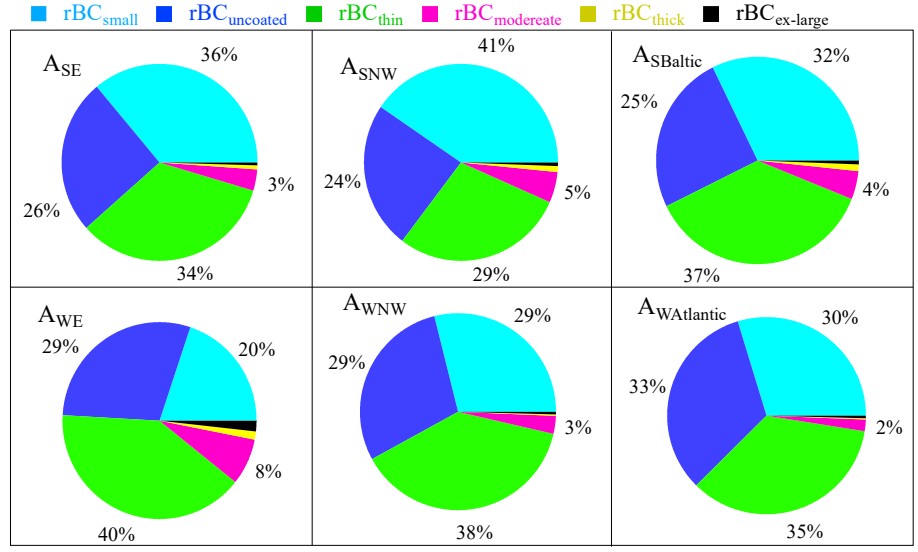

**Figure 12: The rBC core mass fraction of different types of rBC of different air masses in Fig. 10.**

The mechanism of LV-coating formation can be inferred from the diurnal variation of the mixing state of rBC-containing particles in the TD sample. In summer, photochemistry significantly influences the LV-coating as well. As shown in Fig. 13(a), similar to the ambient sample, an increasing volume in rBC$_{modrate}$ and rBC$_{thick}$ region can be observed during



daytime in $A_{SE}$ and $A_{SNW}$. $A_{SBaltic}$ exhibited similar diurnal variation to $A_{SNW}$ (Fig. s5). The LV-coating is primarily associated with the more oxygenated oxidized organic aerosol (MO-OOA), as it exhibited a similar diurnal variation pattern shown in Atabakhsh et al. (2023). Moreover, as shown in Fig. 13(b) and Table.4, the statistical properties of the

mean $mf_{coated}$ in the TD sample were similar across different air masses during summer. However, more particles with thick LV-coatings can be observed in $A_{SNW}$ and $A_{SBaltic}$, as the 75th percentile of the mean CT values were higher during $A_{SNW}$ (46.35nm) and $A_{SBaltic}$ (45.27nm) than during $A_{SE}$ (37.02nm). The difference in mean CT between different air masses in the TD sample is more obvious than in the ambient sample. Combined with more $rBC_{moderate}$ observed in $A_{SNW}$ and $A_{SBaltic}$, it appears that thick LV-coating is more related to the aged rBC than the freshly emitted rBC. Furthermore,

the diurnal pattern of mean CT (Fig. 13 (c)) and $mf_{coated}$ of $A_{SNW}$ and $A_{Sbaltic}$ exhibited similar variations to the size-resolved CT (Fig. 13 (a)), with peak values occurring at noon. However, in $A_{SE}$, the relatively steady increase in mean $mf_{coated}$ during daytime indicated the photochemical process promotes the condensation of LV-coating on freshly emitted rBC from local sources. The consistent increase of the CT during the daytime, particularly accelerating in the afternoon, implies the contribution the stronger solar radiation to further LV-coating growth.

In contrast to summer, the mixing state of rBC in the TD sample did not exhibit evident diurnal variations in winter. As shown in Fig. 13(a), for $A_{WE}$, only the decrease in volume within the $rBC_{moderate}$ and $rBC_{thick}$ region can be observed at rush hours after 15:00, accompanied by a reduction in $rBC_{moderate}$ core mass fraction from 9 % to 6 %. The $A_{WNW}$ and $A_{WAtlantic}$ displayed a similar plot for all day (Fig. s5) without significant variations. In addition, MO-OOA, associated with residential heating emissions, also exhibited similar variation as described in Atabakhsh et al. (2023). Furthermore,

the statistical properties of mean CT and $mf_{coated}$ showed a decreasing trend from $A_{WE}$ to $A_{WAtlantic}$, which is contrary to the temperature variation across the different air masses. In particular, the mean CT ($32.79 \pm 9.06$nm) of $A_{SE}$ is significantly higher than $A_{WNW}$ ($21.64 \pm 4.27$nm) and $A_{WAtlantic}$ ($20.41 \pm 3.63$nm). Compared to the mean CT, the mean $mf_{coated}$ showed an evident diurnal variation for all air masses, with the higher values observed at night and the low values during daytime. Additionally, $A_{WAtlantic}$ exhibited a much more significant diurnal variation compared to $A_{SW}$. This significance of $mf_{coated}$

diurnal variation can be attributed to the ambient temperature of different air masses as well. In warm periods, heating is only required on cold nights, while in colder periods, biomass or coal is burned more continuously. These observations suggested that LV-coating during winter is more related to the emission sources particularly the residential heating, than the atmospheric process.





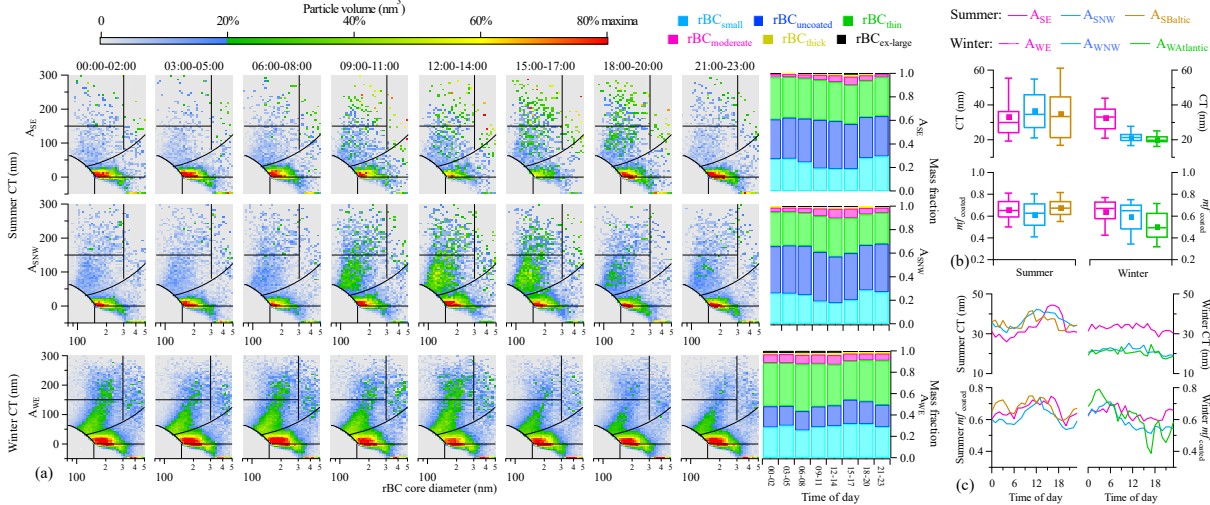

**Figure 13: (a) Diurnal variation of size-resolved coating thickness and mass fraction of each type of rBC in the TD sample during $A_{SE}$, $A_{SNW}$, and $A_{WE}$. (b) statistical analysis and (c) diurnal variation of average CT and $mf_{coated}$ of rBC in the TD sample.**

**Table 4: 25th percentile, 75th percentile, and mean values of $mf_{coated}$ and CT in Fig. 11(b).**

|  |  | Summer | | | Winter | | |
|---|---|---|---|---|---|---|---|
|  |  | $A_{SE}$ | $A_{SNW}$ | $A_{SBaltic}$ | $A_{SE}$ | $A_{SNW}$ | $A_{SBaltic}$ |
| CT (nm) | 25th~75th | 23.68~37.02 | 26.60~46.35 | 20.81~45.27 | 23.68~37.02 | 26.60~46.35 | 20.81~45.27 |
|  | Mean | 33.14 ± 13.72 | 36.69 ± 12.92 | 35.29 ± 15.94 | 33.14 ± 13.72 | 36.69 ± 12.92 | 35.29 ± 15.94 |
| $mf_{coated}$ | 25th~75th | 0.58~0.74 | 0.51~0.72 | 0.61~0.74 | 0.58~0.74 | 0.51~0.72 | 0.61~0.74 |
|  | Mean | 0.66 ± 0.11 | 0.61 ± 0.14 | 0.68 ± 0.09 | 0.66 ± 0.11 | 0.61 ± 0.14 | 0.68 ± 0.09 |

## 4 Conclusion

To investigate the microphysical properties of atmospheric black carbon aerosols, measurements were performed at the central background site Melpitz, Germany in the summer (August 2021) and winter (December 2021) using an SP2. The core size and coating thickness were examined using the single-particle data, and the coating volatility of rBC was investigated by the sample air passing through a TD with a temperature of 300°C upstream of the SP2.

Backward trajectory analysis and wind direction patterns identified one air mass influenced by the easterly winds and two
air masses associated with long-range transport arriving from the west of Melpitz during both seasons. In summer, the air mass corresponding to the easterly winds was more linked to the local emissions from the Melpitz village. The smaller rBC sizes (MMD of 164 nm), combined with the detection limit of the SP2, introduced an overestimation of average CT. Higher fractions of thickly coated rBC (CT>~50nm) and increased fraction of rBC with LV-coating were both observed in the afternoon, suggesting the contribution of photochemical processes to the mixing state of rBC. In winter, the easterly
winds were associated with both local emissions and transportation from Eastern Europe. The residential heating with



biomass burning during winter contributed to the observed largest MMD (216 nm) and thick CT (63.16 ± 12.65 nm) in this air mass.

For the long-transported air masses during summer, the smallest MMD (140 nm) was observed. The highest fraction of thickly coated rBC was found at noon for these long-transported air masses, differing from the local emissions, related to the aged rBC that already acquired coatings during transportation. In winter, the ambient temperature of long-transported air masses from the west was higher than the air mass influenced by the easterly winds, potentially leading to less residential heating emissions. As a result, smaller core sizes and thinner CT were observed in these long-transported air masses. The LV-coating didn't exhibit evident diurnal variation for all air masses during winter. However, the cold air masses exhibited a thicker LV-coating compared to the warm air masses, suggesting the emission sources play a more important role in the coating volatility than the atmospheric processes during winter.

The TD-SP2 system provides a valuable tool for determining the composition of coatings, including the volatile coating (comprised of secondary inorganic components and volatile organic components) and the low-volatile coating (consisting of low-volatile organic components). This study found that at Melpitz, LV-coatings in summer were largely associated with secondary organic formation through photochemical processes, while in winter, they were more related to residential heating emissions. The results obtained from this study will provide insights into aerosol dynamical processes associated with coating formation, and allow the attribution of certain aerosol species that influence the radiative properties of BC.

**Supplement.** The supplement related to this article is available online at: XXX.

**Author contributions**. TM and YY designed the research. YY, LP, and JV collected the data at Melpitz. YY, BH, LP, and JV performed the data analysis. YY wrote the first draft of the manuscript. All co-authors contributed to the interpretation of the results as well as paper review and editing.

**Data availability:** Data are available upon request to the corresponding author.

**Competing interests.** The authors have no competing interests to declare.

**Acknowledgment.** We thank Christopher Pöhlker for the helpful suggestions and comments on improving this paper.

**Financial support.** This research has been supported by the China Scholarship Council (grant no. 201908320539), ACTRIS2 (grant no. 654109), and the TRACE project funded by the CSF (grant no. 20-08304J) and by DFG (grant no. 431895563).



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
