# Peer review of "Microphysical properties of refractory black carbon aerosols for different air masses at a central European background site"

_EGUsphere, 2024_

## Author Comment (AC1)

**RESPONSE LETTER#1 (egusphere-2024-3539)**

Dear Anonymous Referee #1:

We would like to thank the referee for the comments. In the following paragraphs, we report the referee's comment in black and the author's answer in blue. The modification in the manuscript is marked as blue.

The manuscript "Microphysical properties of refractory black carbon aerosols for different air masses at a central European background site" presents a study on black carbon properties from a central European measurement site in December and in August. It focuses on how the air-mass origin influences these properties, including not only BC mass concentration and size distribution but also the BC coating thickness. I think it is a very valuable study, which suits well the scope of the journal, and it is worth publishing, however, the manuscript still needs important changes to be better understandable. I think that many of the discussions are way too long and too specific, tries to explain even the smallest details of the different figures and sometimes even speculate too much what could cause those small details, that might not even be there considering the uncertainty of the measurements. I do think, that less would be more here. I often needed to read sentences twice until I really understood what the authors would like to say, and then I needed to look at the figures for a long time to see that as well. I do suggest focusing more on the important things. I hope my detailed comments help with that.

Our response: We thank the reviewer for the thoughtful and valuable comments and suggestions, which were very helpful in improving our manuscript. We revised the manuscript carefully, as described in our point-to-point responses to the comments.

L33-35: In my opinion, the uncertainty in BC lifetime is also an important factor here, please consider adding it.
Our response: Thanks for your comment. We have added "lifetime" to the text.

L40-41: BC is not hydrophyobic, that would mean a hygroscopicity worse than a non-soluble material, which is to my knowledge not the case. BC is simply non-hygroscopic. Please correct. And I do not really understand what the authors mean by "and independent from other atmospheric materials", please explain better.

Our response: Thanks for your correction. We have modified the text as: "In general, freshly emitted BC exhibits low hygroscopicity and less mixing with other atmospheric materials (Schwarz et al., 2008a; Liu et al., 2013). "

L41: "After emissions" please change it to "After emission"

Our response: We have changed "emissions" to "emission".

L56: start maybe a new paragraph here

Our response: We have started a new paragraph here.

L58-62: sentence too long and hardly understandable. Please make 2 separate sentences out of it and reformulate for better understandability.

Our response: We have split the sentence to:

"These studies assume that the coating of BC is completely removed after passing through the thermodenuder, or they only consider the effect of non-refractory coatings on absorption. However, some low-volatile (LV) coatings remain on BC after heating (Poulain et al., 2014). These components can lead to inaccurate estimates of the lensing effect (Shetty et al., 2021). "

L75: please change "volatility" to "volatile"

Our response: We have changed the "volatility" to "volatile".

L75-77: You have selected the months with the highest and lowest BC concentrations of the year to analyze. Why not the whole seasons or even the whole year if you do have the data? And if those two months are the ones with the highest and the lowest BC concentrations, that means to me that in the other months of the winter the BC concentrations are lower and in summer higher. With that, in my opinion, December and August cannot represent well winter and summer. Please comment on this. As a suggestion, you could say, that you only focus on the two months with the highest and lowest BC concentrations. But still my question remains, why not the other months? Are those not that interesting?

Our response: Thanks for the comments. The reason we selected these two months is based on the rBC properties. As shown in Fig.1, the CT diurnal variation exhibited two different modes: one with a peak during the daytime (September to October) and another one with a peak at night (November to January). The most evident diurnal variations of these two modes, with relatively higher CT, were found in August and December, respectively. In addition, the highest and lowest concentrations and the most significant shift of size distribution were observed in these two months as well, as shown in Fig.2 below. These two months exhibited the most contrasting rBC properties and coincidentally fall within summer and winter. Furthermore, practice constrains our selection as well, including the limited measurement in summer, delays in ACSM data processing, and instrument maintenance.

To clarify these in the manuscript, we have revised the text to:

" In this study, we focused on August and December, the two months with the most contrasting rBC properties, which coincidentally correspond to summer and winter. As shown in Fig. S1 and S2, these months exhibited the highest and lowest BC mass concentrations, the smallest and largest rBC core sizes, and distinct diurnal variations in coating thickness.".

[Figure]

Figure.1 Diurnal variation of rBC mass concentration and coating thickness of different months.

[Figure]

Figure.2 rBC mass size distribution of different months.

L82-83: Figure 1 does not show any measurements, maybe change it to measurement site?

Our response: We have changed the sentence to:

 "Figure 1 shows measurement site Melpitz (12°56'E, 51°32'N, 86 ma.s.l.) of Leibniz Institute for Tropospheric Research (TROPOS) ".

L115-117: please make it clear, if you have just identified the amount of the "missing" BC or used it for correction, and reported the corrected or uncorrected mass concentration values!

Our response: Thanks for your comment. We have revised the sentence for clarity: "In this study, the detection range of this SP2 is 80-500 nm. The missing ratio due to the SP2 detection limit in summer and winter was $17 \pm 7\%$ and $5\pm 4\%$, respectively. After correction, the average rBC mass concentrations increased from $0.13\pm0.11$ μg m$^{-3}$ to $0.16 \pm 0.12$ μg m$^{-3}$ in summer, and from $0.59\pm0.55$ μg m$^{-3}$ to $0.61 \pm 0.56$ μg m$^{-3}$ in winter. "

L118: 4000k, K should be capital letter

Our response: We have changed the "k" to "K".

L132-133: "When considering the average coating thickness, these negative values were not counted" In my opinion the negative values should be considered when calculating the average coatings. Determining Dp with the LEO fit, has a great uncertainty (by the way, I would really like to see a discussion on that and the extent of that). And as you mention this is also one of the main reasons why negative CT values are present. I believe that this uncertainty results with kind of the same probability in a too large and a too small Dp. And with this only ignoring the

negative CT values, you simply ignore a fraction of values where a too small Dp was determined. And this results in artificially too high averages. The case is of course different if the negative CT originates from the refractive index values not perfectly representing the real values. However, that gives you an uncertainty that you cannot get rid of with ignoring the negative CT values, because that bias is present than for all particles. So that would also not make it useful to ignore the negative CT values. The only case where ignoring the negative CT values would make sense is maybe the case where the core is not in the middle of the particle. But I guess that is only a minor fraction?

Our response: Thanks for your correction. We acknowledge that excluding negative CT values will remove the smaller $D_p$ values, thus biasing the mean CT toward higher values. To address this, we re-calculated the mean CT by considering the negative values. As shown in Fig. 4 below, this adjustment didn't significantly affect the average CT, and the overall variation trend remained consistent. Consequently, our main conclusions remain unaffected. We have also updated the corresponding text and figures correlated to the mean CT accordingly.

In addition, if we think of the negative CT as the uncertainties of LEO-fit, we can not determine whether the rBC particles with negative CT are uncoated. To avoid potential misunderstanding, we have removed the discussion correlated to the uncoated rBC and mass fraction of coated rBC ($mf_{coated}$). In the size-resolved CT, we have counted the $rBC_{uncoated}$ into $rBC_{thin}$.

[Figure]

Fig.3 Time series of average CT

We have also revised the description of negative CT as follows:

" These negative coatings can be attributed to the uncertainties of LEO-fit, which resulted from several factors such as the noise of both scattering and incandescent signals, calibration of SP2, and the core position of the particle (Laborde et al., 2013; Taylor et al., 2015; Krasowsky et al., 2018; Ko et al., 2020). When considering a large number of particles, Taylor et al. (2015) found

the systemic uncertainty of the averaged particle population is primarily influenced by the assumption of rBC core refractive index and density. These systemic errors do not affect comparisons within measurements using the same set of parameters assumed for the rBC (Ko et al., 2020). Laborde et al. (2013) also noted these systemic errors are less concerning when averaging over a large particle population. These considerations justify the inclusion of negative CT values in the calculation of the mean. "

L143-144: people outside of the SP2 community might not understand this factor of 0.75, please add one or two more sentences to explain that.

Our response: We have added the sentences to the text:

"The SP2 exhibits a higher sensitivity to Aquadag particles compared to ambient soot. For the same mass, Aquadag generates a stronger incandescent signal. Therefore, a correction factor of 0.75, as suggested by Baumgardner et al. (2012) and Laborde et al. (2012), was applied to the incandescent signal during calibration."

L152: "compounds(Poulain et al., 2014)" space is missing

Our response: We have added the space.

Figure 2 caption: "Instruments setup" please change it to "Instrument setup"

Our response: We have changed the "Instruments setup" to "Instrument setup".

Line175: "during summer and winter were shown in Fig. 3." Please change "were" to "are"

Our response: We have changed "were" to "are"

Figure3: the labels of the AMS measurements showing the different components are very small, maybe make them a bit larger

Our response: We have enlarged the labels.

Figure3 caption: "mean mass fraction (mfcoated)" please indicate that you show here the mass fraction of coated particles

Our response: Thanks for your comment. Based on the previous comment, we have removed the discussion on $mf_{coated}$ from the manuscript.

Figure 3 caption: "The shaded areas indicate air masses." Change it maybe to "The shaded areas indicate different air masses."

Our response: We have changed the "The shaded areas indicate air masses" to "The shaded areas indicate different air masses. "

Table 1: "mean values and variability" is the variability you show there the standard deviation? Also everywhere else in the paper? Please state.

Our response: The term "variability" here refers to the range of variation in these parameters. We have revised "mean values and variability" to "range of variation and mean values. " Additionally, we have clarified that the value following "±" represents the standard deviation when it first appears in line 176.

Table 1: it is strange to show a range with a "~" sign. Please consider changing it to "-". Does the ranges correspond to the minimal and maximal values of the 1-hour averages? Please state as well.

Our response: We have changed "~" to "-", and added the description "The ranges correspond to the minimal and maximal values of the 1-hour averages" to the caption.

Table 1: "25.32~8. 28" Typo, please change ~ to +-.

Our response: We have changed "~" to "±".

L180-182: was the MMD fitting done on the 1-hour average size distributions? Please mention.

Our response: We have mentioned that the MMD is from the 1-hour average size distributions.

L184: remove the additional "," at the end of the sentence.

Our response: We have removed the ",".

L190-192: the mixing state of the particles is not discussed in the "next chapter" but in chapter 3.3. the one after. Please correct.

Our response: We have changed the "next chapter" to "later".

L201-203: it is the first time here, that back trajectories, clustering and NWP was mentioned. Please add a section to the methods, and introduce what methods were used and how. Adding only one reference and no more details is not enough in my opinion.

Our response: We have added a new chapter in the "Method" part:

**2.3 Non-parametric Wind Regression and Backward Trajectory Analysis**

To investigate both the local and predominant wind sector associated with transported emission sources and rBC properties, we performed a Non-parametric Wind Regression (NWR) analysis using ZeFir, an Igor-based tool developed by Petit et al. (2017). NWR smooths data over a fine grid, allowing the estimation of weighted concentrations for any wind direction ($\varphi$) and wind

speed (v) pair, with weighting coefficients determined via Gaussian-like functions (Henry et al., 2009). Additionally, we employed the NOAA HYbrid Single-Particle Lagrangian Integrated Trajectory (HYSPLIT-4) model to generate 72-hour backward trajectories at 1-hour resolution, at 100 m above the sampling site's ground level. These trajectories trace air parcel origins and transport pathways, providing insight into potential pollutant source regions (Cohen et al., 2015). The resulting backward trajectories are presented in Fig. S3. Furthermore, to identify periods with similar geographical source regions and rBC physical properties, the cluster analysis was subsequently applied to the backward trajectories by using ZeFir. The optimal number of clusters was determined based on the total spatial variance (TSV) (Syakur et al., 2018) and three different clusters were identified in each season.

L203: "measured above 6 meters above the ground." Please remove the first "above"

Our response: We have removed the first "above".

L205: not in Table 4, in Table 2, please change.

Our response: We have changed "Table 4" to "Table 2".

L215: "All trajectory clusters arrived in the northwestern or southwestern sections of Melpitz"
Is A_SBaltic is not north-east?

Our response: Thanks for your comment. The $A_{SBaltic}$ cluster originated from the northeast of Melpitz. However, as shown in Fig. 4(a) of the manuscript, it changes direction in northeastern Germany before ultimately arriving at the northwestern section of Melpitz.

L201-214: Add a sentence that you will later define AWE and ASE.

Our response: We have added the sentence at L205:

"As shown in Fig.4 (a) and (b), three different clusters of back trajectories were classified, corresponding to $A_{SNW}$ and $A_{SBaltic}$ in summer and $A_{WNW}$ and $A_{WAtlantic}$ in winter. $A_{SE}$ and $A_{WE}$ will be defined in the following paragraph. "

L240-241: Add a reference to Fig 5c

Our response: We have added the reference to Fig.5c.

L262: "sizes(Liu et al., 2014; Zhang et al., 2020)" space missing

Our response: We have added the space.

L293-294: I do not understand this sentence, please rewrite, do you mean here, that the ambient samples were considered for the mixing state analysis and not the thermodenuded ones?

Our response: We have changed the sentence to:

"In this chapter, we focus on the mixing state of rBC in the ambient sample. The analysis of the TD sample will be presented in the next chapter."

L295-299: why volume? Why not simply the number of BC particles?

Our response: Thanks for your comment. Due to the small MMD of rBC at Melpitz, there are significantly more small rBC cores than large ones. A substantial fraction of coated rBC falls below the detection limit, preventing us from obtaining coating information. As shown in Fig.4 below, when using particle numbers, there are no significant differences in the size-reolved CT patterns across different air masses. Since rBC optical properties are correlated with particle volume, where rBC with thicker coatings or larger cores absorb more light, and inspired by Liu et al. (2014), who used the scattering enhancement of coated rBC (which is volume-dependent) to quantify the size-resolved mixing state and perform source apportionment, we decided to use particle volume for our analysis. This approach allows for a more distinct characterization of the size-resolved mixing state across different air masses.

[Figure]

Fig.4 Size resolved CT quantified by rBC numbers of different air masses.

L300-302: "The area labeled 'Small rBC without coating information' (rBCsmall), as shown in the bottom-left slashed region of each panel of Fig. 7, indicates rBC particles that exhibited neither a positive CT nor a negative CT." ??? Is this not the category, where simply Dp is too small to determine with the LEO-fit, and therefore you do not have any data on their coating?

Our response: Thanks for your comment. Based on the later comment, no further changes are needed here.

Figure 7 caption: the black dashed lines, I do not see y=300 line. Did you not mean x=300 there? And there are more lines than equations. Please check that again.

Our response: Thanks for your correction. We have changed the description to:

"The vertical dash lines from left to right are x=150 and x=300 respectively. The horizontal dash lines from top to bottom are y=150, y=0.25x, y=140.5-0.89x".

L308: "most particles exhibited negative (rBCsmall)" Is that not then the category rBCuncoated and not rBCsmall?

Our response: Thank you for your comment. Based on previous feedback, we have removed the discussion on $rBC_{uncoated}$ from the manuscript.

L311-314: This explanation should come where you define the rBCsmall, and then you can ignore my comment for L300-302.

Our response: Thanks for your comment.

L315: it is strange here as well to use the "~" sign to present ranges. Please consider using "-" here (and everywhere) as well.

Our response: We have used "-" to present ranges instead of " -".

L316-317: please change "associated with the smaller size distribution" it to "associated with size distribution shifted towards smaller sizes"

Our response: We have changed "associated with the smaller size distribution" to "associated with size distribution shifted towards smaller sizes".

L317-320: very good that you discovered, and describe this problem of not being able to detect the thin coating of the small BC particles, but do you also have an estimate of how much your CT values are biased due to it? If not, would that not be better just to not include any CT values in e.g. the average CT values for Dc below 150? And state that clearly?

Our response: Thanks for your question. Estimating the extent of bias in CT values due to the detection limit is challenging, as we cannot directly assess the missing coating information or measure the coated rBC size distribution. Currently, there is no standardized approach within the SP2 community for selecting a specific core size range when calculating the average CT. In this study, we calculated the mean CT across the full detection range of rBC cores because one of our objectives was to evaluate whether this full-range mean CT effectively captures variations in rBC mixing state. Additionally, CT values for rBC cores smaller than 150 nm exhibited distinct characteristics across different air masses. Given that rBC core size distributions can shift toward smaller sizes, it is reasonable to include these values in the overall analysis. Therefore, we decided to retain the average CT calculated for the 80–500 nm range.

L357-358: why are heating systems running continuously in Melpitz in August? Are you sure, that this is the source for the thinly coated BC throughout the day?

Our response: The heating system is not only for the house warming purpose but also for the water heating. To clarify this point and avoid confusion, we have revised the text by specifying " (housewarming and water heating) " after mentioning the heating system the first time.

L358-360: "However, during evening rush hours after 18:00, the increasing volume in rBCthin and rBCuncoated regions can be found in the size-resolved CT plot forASE and ASNW." I do see the increase in rBCuncoated, but not at all in rBCthin. You also state in the previous sentence that there is not really a diurnal variability in rBCthin. I have the feeling that you try to explain even the smallest changes and speculate a bit too much about this figure. If I look at it, I simply see, that the diurnal behaviour is very similar for all airmasses in summer, and there is almost no diurnal variability, the only thing which might be there is that there are a bit higher fractions of uncoated particles during the rush hours, and the amount of moderately and thickly coated during the day. I would not overcomplicate the description of this plot.

L373-385: same for the winter description, that could also be made a bit simpler. And I also do not see everything that is stated there. E.g. "In contrast, high volumes of rBCmoderate or rBCthick were observed at night for AWE and AWNW…" I do see higher volumes of rBCmoderrate from somewhere 3:00 to 12:00, that is not exactly night for me. Please rewrite a bit this part. And another suggestion, or idea: it seems that the most important influence is the local influence on the diurnal variation of CT. Would that not be simpler, if you would not sort this data according to the different air-masses, and just show and discuss the diurnal variability for the combined data? Just an idea.

Our response: Thank you for your constructive suggestions. Since these two comments address the same context, we have combined our responses. We agreed that we previously over-analyzed the small details of the plot and tried to explain too much. We have combined the data and moved the diurnal variation of different air masses in the Supplement, and removed the discussion and plot related to the $mf_{coated}$. The revised text is as follows:

" Figure 9 displays the diurnal variation of size-resolved coating thickness of the overall data in both seasons. The different masses exhibited similar diurnal patterns within each season as shown in Fig. s5. During summer, a high volume (red) of $rBC_{thin}$ can be observed most of the day. This observation is associated with the liquid fuel combustion of the continuously running heating system of Melpitz village (van Pinxteren et al., 2024). Throughout the daytime from 09:00 to 18:00, a continuous increase in particle volume in the $rBC_{moderate}$ and $rBC_{thick}$ region is observed, with the highest total mass fraction of $rBC_{moderate}$ and $rBC_{thick}$ reaching a maximum of ~22 % between 12:00 and 15:00. This observed transition to the thicker coating region can be regarded as evidence of coating growth of rBC-containing particles, implying a significant

role of photochemical reactions in the aging process of rBC, which has been reported in other studies (Liu et al., 2014; Yang et al., 2019). Later in the evening rush hours after 18:00, an increase in the volume of $rBC_{thin}$ regions and the core mass fraction of $rBC_{thin}$ was observed. This is inconsistent with the mass concentration peak of rBC shown in Fig. 5, which correlated to the traffic emissions.

The temperature can be considered a crucial factor influencing the mixing state of rBC at Melpitz in winter, giving the impact on the residential heating emissions. In contrast to summer, the high volumes of $rBC_{moderate}$ and $rBC_{thick}$ were observed throughout the day during winter, except between 15:00 and 21:00, when their core mass fraction decreased to a minimum of 16%. This reduction correlated to the reduced residential emissions due to the higher temperature in the daytime and the increased traffic related $rBC_{thin}$ during rush hour. Between 00:00 to 06:00, the relatively higher (maximum of 26%) core mass fraction of $rBC_{moderate}$ and $rBC_{thick}$ was observed, coinciding with the lowest temperatures of the day (Fig. S6) with increased residential heating emissions. "

[Figure]

**Figure 9: Diurnal variation of size-resolved coating thickness and mass fraction of each type of rBC in the ambient sample. The panels for each season, from left to right, represent time intervals of 00:00–03:00, 03:00–06:00, 06:00–09:00, 09:00–12:00, 12:00–15:00, 15:00–18:00, 18:00–21:00, and 21:00–00:00. The color scale represents the total particle volume, with red set at 80% of the maximum value.**

[Figure]

**Figure 10: (a) Statistical analysis and (b) diurnal variation of average CT of rBC in the ambient sample. (c) Diurnal variation of chemical composition mass concentration.**

Additionally, in order to maintain consistency in the manuscript, we also adjusted Fig.13 and split it into two figures to ensure structural coherence. The corresponding text has also made some minor changes.

[Figure]

**Figure 13: Diurnal variation of size-resolved coating thickness and mass fraction of each type of rBC in the TD sample.**

[Figure]

**Figure 14: Statistical analysis and diurnal variation of CT for each air mass in winter and summer.**

Figure 9: The times seem strange for the diurnal categories. 00-02, then 03-05, then 06-08… There is always an hour missing. Was the data really sorted like that? Why? Or is that only a typo?

Our response: This is the typo, thanks for your correction. We have changed the time ranges to "00-03, 03-06, 06-09…."

Figure 10: The discussion about this figure would be better right after the discussion on figure 5. Here, I had to go back and forth between figure 5 and 10 to follow it. Please consider it. And again, the discussion seems to me here as well overcomplicated. I would discuss this right after Figure 5 and state that in the average CT there is no real difference in summer between the air-masses, whereas in winter there is. And then after this discuss the size resolved CT figures. Showing what you see there, and also what more that brings compared to the bulk CT.

Our response: We appreciate your suggestion to move the discussion of Figure 10 closer to Figure 5 to improve readability. However, we have decided to maintain the current structure to present the rBC properties separately in a logical sequence —the mass concentration, size distribution, mixing state, and coating volatility. This structure would avoid discussing the mixing state in different sections. In addition, we first analysed the size-resolved CT to address the impact of the detection limit and better investigate the mixing state of rBC. We then compared the bulk CT with size-resolved CT to assess whether the bulk CT adequately captures

variations in the mixing state. Furthermore, the comparison between Figures 5 and 10 is minimal, with the only common point being the discussion of rush hour related to the thinly coated or uncoated rBC. For these reasons, we believe the existing structure is more effective and would prefer to keep it unchanged.

L396-397: "However, the 75th percentile CT in ASE is lower than the other two air masses during summer" and the 25th percentile is higher. If you discuss why the 75th percentile is lower, then you should discuss why the 25th is higher as well. Again, here, probably speculating less would be better.

Our response: Thanks for your comment. In line with your previous suggestion to simplify the discussion, we have removed this discussion, along with similar discussions in Section 4.3, to avoid overcomplicating the interpretation of the results.

L398: "In winter, the influence of the detection limit seems to be less significant." What do you base this statement on? Please explain. I do not understand it.

Our response: Thank you for your comment. In winter, the rBC core size distribution shifts toward larger sizes, resulting in fewer small particles falling below the detection limit. However, we cannot estimate how the size distribution affects the detection limit. To avoid any misleading implications, we have removed this sentence from the text.

L419: "studies(Xu et al., 2018; Wang et al., 2021a)" space is missing

Our response: We have added the space.

Figure 11 (and same for Figure 8): the blue color of the pie chart does not match the blue of the label, please change.

Our response: We have changed the color.

L439-440: "although the reduced core mass fraction of rBCthin in the TD sample ranged from 8 % to 13 % compared to the ambient sample" I think you do want to write here, that the masss fraction was 8-13% less of rBCthin after the thermodenuder compared to the ambient case. As it is written now, I do not understand. Please reformulate.

Our response: We have changed this sentence to: "mass fraction of $rBC_{thin}$ in the TD sample was reduced by 8–13% compared to the ambient sample"

L465: "In contrast to summer, the mixing state of rBC in the TD sample did not exhibit evident diurnal variations in winter" What about the diurnal variation of mfcoated? There I do see a diurnal variation.

Our response: Thanks for the comments. In this context, "mixing state" specifically referred to size-resolved CT. To avoid misunderstanding, we have changed the "mixing state" to "size-resolved-CT".

**References:**

Cohen, M. D., Stunder, B. J. B., Rolph, G. D., Draxler, R. R., Stein, A. F., and Ngan, F.: NOAA's HYSPLIT Atmospheric Transport and Dispersion Modeling System, Bulletin of the American Meteorological Society, 96, 2059-2077, 10.1175/bams-d-14-00110.1, 2015.

Henry, R., Norris, G. A., Vedantham, R., and Turner, J. R.: Source Region Identification Using Kernel Smoothing, Environmental Science & Technology, 43, 4090-4097, 10.1021/es8011723, 2009.

Ko, J., Krasowsky, T., and Ban-Weiss, G.: Measurements to determine the mixing state of black carbon emitted from the 2017–2018 California wildfires and urban Los Angeles, Atmospheric Chemistry and Physics, 20, 15635-15664, 10.5194/acp-20-15635-2020, 2020.

Krasowsky, T. S., McMeeking, G. R., Sioutas, C., and Ban-Weiss, G.: Characterizing the evolution of physical properties and mixing state of black carbon particles: from near a major highway to the broader urban plume in Los Angeles, Atmospheric Chemistry and Physics, 18, 11991-12010, 10.5194/acp-18-11991-2018, 2018.

Laborde, M., Crippa, M., Tritscher, T., Jurányi, Z., Decarlo, P. F., Temime-Roussel, B., Marchand, N., Eckhardt, S., Stohl, A., Baltensperger, U., Prévôt, A. S. H., Weingartner, E., and Gysel, M.: Black carbon physical properties and mixing state in the European megacity Paris, Atmospheric Chemistry and Physics, 13, 5831-5856, 10.5194/acp-13-5831-2013, 2013.

Liu, D., Allan, J. D., Young, D. E., Coe, H., Beddows, D., Fleming, Z. L., Flynn, M. J., Gallagher, M. W., Harrison, R. M., Lee, J., Prevot, A. S. H., Taylor, J. W., Yin, J., Williams, P. I., and Zotter, P.: Size distribution, mixing state and source apportionment of black carbon aerosol in London during wintertime, Atmospheric Chemistry and Physics, 14, 10061-10084, 10.5194/acp-14-10061-2014, 2014.

Petit, J. E., Favez, O., Albinet, A., and Canonaco, F.: A user-friendly tool for comprehensive evaluation of the geographical origins of atmospheric pollution: Wind and trajectory analyses, Environmental Modelling & Software, 88, 183-187, 10.1016/j.envsoft.2016.11.022, 2017.

Syakur, M. A., Khotimah, B. K., Rochman, E. M. S., and Satoto, B. D.: Integration K-Means Clustering Method and Elbow Method For Identification of The Best Customer Profile Cluster, IOP Conference Series: Materials Science and Engineering, 336, 012017, 10.1088/1757-899X/336/1/012017, 2018.

Taylor, J. W., Allan, J. D., Liu, D., Flynn, M., Weber, R., Zhang, X., Lefer, B. L., Grossberg, N., Flynn, J., and Coe, H.: Assessment of the sensitivity of core / shell parameters derived using the single-particle soot photometer to density and refractive index, Atmospheric Measurement Techniques, 8, 1701-1718, 10.5194/amt-8-1701-2015, 2015.

---

## Author Comment (AC2)

**RESPONSE LETTER#2 (egusphere-2024-3539)**

Dear Anonymous Referee #2:

We would like to thank the referee for the comments. In the following paragraphs, we report the referee's comment in black and the author's answer in blue. The modification in the manuscript is marked as blue.

This manuscript presents a comprehensive analysis of the microphysical properties of refractory black carbon (rBC) aerosols, focusing on their seasonal variability and correlations with air mass trajectories at a central European background site. The data collected, along with the advanced instrumentation employed (e.g., SP2 and thermodenuder systems), provide valuable insights into the physical characteristics and atmospheric interactions of rBC. Overall, the manuscript is well-structured. However, there are a few areas where additional clarification or detail could enhance the clarity of the work.

Our response: We thank the reviewer for the thoughtful and valuable comments and suggestions, which were very helpful in improving our manuscript. We revised the manuscript carefully, as described in our point-to-point responses to the comments.

L75: Why were August and December specifically chosen to represent summer and winter? Are there other reasons for this selection, beyond the observed highest and lowest rBC concentrations?

Our response: Thanks for the comments. The reason we selected these two months is based on the rBC properties. As shown in Fig.1, the CT diurnal variation exhibited two different modes: one with a peak during the daytime (September to October) and another one with a peak at night (November to January). The most evident diurnal variations of these two modes, with relatively higher CT, were found in August and December, respectively. In addition, the highest and lowest concentrations and the most significant shift of size distribution were observed in these two months as well, as shown in Fig.2 below. These two months exhibited the most contrasting rBC properties and coincidentally fall within summer and winter. Furthermore, practice constrains our selection as well, including the limited measurement in summer, delays in ACSM data processing, and instrument maintenance.

To clarify these in the manuscript, we have revised the text to :

"In this study, we focused on August and December, the two months with the most contrasting rBC properties, which coincidentally correspond to summer and winter. As shown in Fig. S1 and S2, these months exhibited the highest and lowest BC mass concentrations, the smallest and largest rBC core sizes, and distinct diurnal variations in coating thickness."

[Figure]

Figure.1 Diurnal variation of rBC mass concentration and coating thickness of different months.

[Figure]

Figure.2 rBC mass size distribution of different months.

Lines 9-10: I suggest revising to: "Uncertainties persist in estimating the radiative forcing of black carbon (BC) due to an incomplete understanding of its microphysical properties."

Our response: Thanks for your comment. We have revised the sentence to "Uncertainties persist in estimating the radiative forcing of black carbon (BC) due to an incomplete understanding of its microphysical properties. "

Lines 82-83: I recommend revising to: "Figure 1 presents measurements taken at the Melpitz research site (12°56'E, 51°32'N, 86 m a.s.l.) of the Leibniz Institute for Tropospheric Research (TROPOS), located 50 km northeast of Leipzig, Germany."

Our response: Thanks for your comment. We have revised the sentence in accordance with the Referee#1's comment, which aligns closely with your recommendation.

L120: Consider adding a brief explanation of the 'LEO-fit' method for readers who may be unfamiliar with this technique.

Our response: We have added the explanation of 'LEO-fit' as follows:

"The leading edge-only (LEO) fit method, which uses a Gaussian fit of the scattering profile before coating evaporation to reconstruct the scattering signal, was applied in this study. Technical details about the LEO-fit approach can be found in Gao et al. (2007). "

Lines 165-168: I suggest revising to: "An aerosol chemical speciation monitor (ACSM, Aerodyne Research, MA, USA; Ng et al., 2011) and an Aerodyne high-resolution time-of-flight aerosol mass spectrometer (HR-ToF-AMS, hereafter referred to as AMS, DeCarlo et al., 2006) were used to measure the bulk chemical composition of non-refractory PM1 aerosol species, including organic aerosols (OA), nitrate, sulfate, ammonium, and chloride."

Our response: We have revised the sentence to "An aerosol chemical speciation monitor (ACSM, Aerodyne Research, MA, USA; Ng et al., 2011) and an Aerodyne high-resolution time-of-flight aerosol mass spectrometer (HR-ToF-AMS, hereafter referred to as AMS, DeCarlo et al., 2006) were used to measure the bulk chemical composition of non-refractory PM1 aerosol species, including organic aerosols (OA), nitrate, sulfate, ammonium, and chloride."

Lines 175-178: I recommend revising to: "Aerosol components exhibited clear seasonal variations. In summer, organic aerosols (OA) dominated, with a mean mass fraction of $55 \pm 13\%$. In contrast, during winter, the OA fraction decreased to $29 \pm 14\%$, while the nitrate fraction significantly increased to $29 \pm 15\%$, compared to $8 \pm 5\%$ in summer." It is important to ensure consistency in the number of decimal places throughout the manuscript.

Additionally, while organic aerosols are the predominant component in summer, nitrate plays a key role in winter. However, the manuscript does not discuss the reasons behind these seasonal variations. I would suggest expanding the discussion on the possible drivers behind

these variations. For instance, could meteorological factors like temperature and humidity be influencing the observed seasonal trends, or are they more likely related to anthropogenic activities? This would provide a more comprehensive context for understanding the data."

Our response: Thanks for your comment. We have received the sentence to "Aerosol components exhibited clear seasonal variations. In summer, organic aerosols (OA) dominated, with a mean mass fraction of $55 \pm 13\%$. In contrast, during winter, the OA fraction decreased to $29 \pm 14\%$, while the nitrate fraction significantly increased to $29 \pm 15\%$, compared to $8 \pm 5\%$ in summer."

We appreciate your suggestion to expand the discussion on the drivers of these seasonal variations. Our manuscript already includes some discussions on these topics. For example, regarding the influence of anthropogenic activities, we have the discussion on the heating system in Melpitz village and residential heating emissions. Additionally, meteorological factors are considered in the discussion of the diurnal variation of the mixing state. While a more detailed and comprehensive investigation would be valuable, it goes beyond the scope of this manuscript and will be explored in future research.

L200: Why were 100-meter back trajectories chosen for the analysis? A brief explanation of this choice and the methodology used for back-trajectory and wind analysis would be beneficial in the methods section.

Our response: Thanks for your comment. We selected the 100-meter back trajectories for analysis because there was no significant difference between the back trajectories at 100 m, 500 m, and 1000 m. As shown in Figure 1, the clusters of back trajectories at different heights followed similar paths and originated from similar directions. Using the lowest altitude ensures the most direct connection to ground-based measurements and aligns well with our wind analysis without losing significant information. Given that higher-altitude trajectories exhibited similar patterns, we found 100 m to be the most relevant choice for assessing local and regional influences on rBC. In addition, we have added a new chapter about the HYSLPIT mode and NWR at the "Method" part:

**2.3 Non-parametric Wind Regression and Backward Trajectory Analysis**

To investigate both the local and predominant wind sector associated with transported emission sources and rBC properties, we performed a Non-parametric Wind Regression (NWR) analysis using ZeFir, an Igor-based tool developed by Petit et al. (2017). NWR smooths data over a fine grid, allowing the estimation of weighted concentrations for any wind direction ($\varphi$) and wind speed ($v$) pair, with weighting coefficients determined via Gaussian-like functions (Henry et al.,

2009). Additionally, we employed the NOAA HYbrid Single-Particle Lagrangian Integrated Trajectory (HYSPLIT-4) model to generate 72-hour backward trajectories at 1-hour resolution, at 100 m above the sampling site's ground level. These trajectories trace air parcel origins and transport pathways, providing insight into potential pollutant source regions (Cohen et al., 2015). The resulting backward trajectories are presented in Fig. S3. Furthermore, to identify periods with similar geographical source regions and rBC physical properties, the cluster analysis was subsequently applied to the backward trajectories by using ZeFir. The optimal number of clusters was determined based on the total spatial variance (TSV) (Syakur et al., 2018) and three different clusters were identified in each season.

[Figure]

Fig.1 Back trajectory clusters of different height during summer and winter.

Lines 267-268: I recommend providing some evidence regarding local residential wood heating or burning activities in this context. Additionally, consider reorganizing this section for better clarity.

Our response: We have added the evidence regarding the residential heating and revised the sentence as follows:

"In addition, van Pinxteren et al. (2024) observed that the fraction of biomass and coal combustion emissions at Melpitz was highest during winter. Similarly, Atabakhsh et al. (2023) also found 85% of BC at Melpitz emitted from biomass and coal combustion during winter.

Therefore, residential heating could be an important emission source correlated to the large rBC core at Melpitz during this season."

L245-260: The role of biomass burning in influencing rBC size is well addressed. However, do you have any hypotheses regarding why rBC size is smaller in summer, aside from biomass burning? Could other emission sources or atmospheric processes also contribute to this seasonal variation?

Our response: Thanks for your comment. This is an important question; however, a definitive answer is beyond the scope of our manuscript. The rBC core is primarily influenced by different emission sources (Bond et al., 2013; Liu et al., 2020). The smaller size of rBC may be attributed to the lowest emission of biomass burning and coal combustion (Atabakhsh et al., 2023) in summer. In addition, the lower fraction of local emission (18%) of local emission compared to winter (34%) could also be one reason for the small size of rBC core; the large rBC could be removed during the long transportation. Furthermore, wet deposition due to precipitation and the activation of rBC as cloud condensation nuclei (CCN) could also contribute to the variation of rBC size. However, further measurements and analyses are needed to fully investigate these mechanisms..

Figure 7: Could you provide the rationale for using particle volume in this plot? A brief explanation would be helpful for clarity.

Our response: Thanks for your comment. Due to the small MMD of rBC at Melpitz, there are significantly more small rBC cores than large ones. A substantial fraction of coated rBC falls below the detection limit, preventing us from obtaining coating information. As shown in Fig.5 below, when using particle numbers, there are no significant differences in the size-reolved CT patterns across different air masses. Since rBC optical properties are correlated with particle volume, where rBC with thicker coatings or larger cores absorb more light, and inspired by Liu et al. (2014), who used the scattering enhancement of coated rBC (which is volume-dependent) to quantify the size-resolved mixing state and perform source apportionment, we decided to use particle volume for our analysis. This approach allows for a more distinct characterization of the size-resolved mixing state across different air masses.

[Figure]

Fig.5 Size resolved CT quantified by rBC numbers of di

L325: Could you provide further evidence or examples to support the connection between "thinly coated rBC" and liquid fuel combustion?

Our response: Thanks for your question. As shown in the figure below from Liu et al. (2019), rBC from traffic emission exhibits a smaller core size and low scattering enhancement compared to rBC from the traffic and wood burning emission, which has a large core size and higher absorption enhancement. The rBC_{thin} in our study exhibited similar properties to those of rBC from traffic emissions. In addition, the mass fraction of rBC_{thin} increased during rush hours when the rBC mass concentration increased, as discussed in Section 3.3.2 of the manuscript. However, more direct evidence would require a source apportionment analysis, such as examining hydrocarbon-like organic aerosols (HOA), which we are unable to conduct currently. This could be a valuable direction in our later research, to combine the source apportionment of rBC with the size-resolved CT.

[Figure]

**Figure 9.** Scattering enhancement ($E_{sca}$) as a function of BC core size ($D_c$) for the three periods (as period I–III indicated in Fig. 3) in Beijing winter (**a–c**), Beijing summer (**d**), London with mixed sources (**e**), and London with traffic source (**f**). Each plot is coloured by particle number density (the colour scale is set to be red when the number density is above 70 % of the maxima in each panel). The particles are separated as four groups using the borders (from top to bottom) at $y = 3.38 + 0.000436* \times \char`^ 5.7$, $y = 2.1$, $x = 0.18$, as shown by dashed lines on each plot. The dashed grey lines on (**a**) denote coating thicknesses mapped on the $E_{sca} - D_c$ plot.

L410-415: The statement "This non-simultaneous variation between CT and mfcoated may relate to local emissions and atmospheric processes" could benefit from further clarification. Could you elaborate on how these factors might contribute to this variation?

Our response: Thanks for your comment. According to the comment of Referee#1, we have removed the content related to $mf_{coated}$ in the manuscript.

An additional suggestion would be the manuscript could benefit from a more detailed explanation of the observed relationship between rBC size and/or coatings with atmospheric chemical processes. Could the authors elaborate on how factors like photochemistry (or indicator), relative humidity, or long-range transport of pollutants might contribute to the observed changes in rBC physical properties?

Our response: We appreciate your suggestion to combine the relationship between rBC properties with atmospheric chemical processes. In our manuscript, we have discussed some of these factors, such as photochemical processes influencing coating growth and volatility during summer, and long-range transport contributing to smaller rBC core sizes, thicker coatings, and a higher fraction of low-volatility coatings. We agree that a more in-depth discussion of chemical processes would enhance the manuscript and provide further insight into rBC properties. A comprehensive analysis of these factors requires additional measurements and data. For instance, investigating photochemical influences in detail would need a thorough analysis of ACSM data, which was not yet available when this manuscript was prepared. Similarly, a systematic assessment of meteorological factors and long-range transport effects,

combined with rBC properties, would extend beyond the content of a single study. This manuscript aims to provide a fundamental characterization of rBC at Melpitz. Follow-up studies will address these additional factors in more detail, including the role of chemical processes and optical properties.

**References:**

Atabakhsh, S., Poulain, L., Chen, G., Canonaco, F., Prévôt, A. S. H., Pöhlker, M., Wiedensohler, A., and Herrmann, H.: A 1-year aerosol chemical speciation monitor (ACSM) source analysis of organic aerosol particle contributions from anthropogenic sources after long-range transport at the TROPOS research station Melpitz, Atmos. Chem. Phys., 23, 6963-6988, 10.5194/acp-23-6963-2023, 2023.

Bond, T. C., Doherty, S. J., Fahey, D. W., Forster, P. M., Berntsen, T., DeAngelo, B. J., Flanner, M. G., Ghan, S., Kärcher, B., Koch, D., Kinne, S., Kondo, Y., Quinn, P. K., Sarofim, M. C., Schultz, M. G., Schulz, M., Venkataraman, C., Zhang, H., Zhang, S., Bellouin, N., Guttikunda, S. K., Hopke, P. K., Jacobson, M. Z., Kaiser, J. W., Klimont, Z., Lohmann, U., Schwarz, J. P., Shindell, D., Storelvmo, T., Warren, S. G., and Zender, C. S.: Bounding the role of black carbon in the climate system: A scientific assessment, Journal of Geophysical Research: Atmospheres, 118, 5380-5552, 10.1002/jgrd.50171, 2013.

Cohen, M. D., Stunder, B. J. B., Rolph, G. D., Draxler, R. R., Stein, A. F., and Ngan, F.: NOAA's HYSPLIT Atmospheric Transport and Dispersion Modeling System, Bulletin of the American Meteorological Society, 96, 2059-2077, 10.1175/bams-d-14-00110.1, 2015.

Henry, R., Norris, G. A., Vedantham, R., and Turner, J. R.: Source Region Identification Using Kernel Smoothing, Environmental Science & Technology, 43, 4090-4097, 10.1021/es8011723, 2009.

Liu, D., He, C., Schwarz, J. P., and Wang, X.: Lifecycle of light-absorbing carbonaceous aerosols in the atmosphere, npj Climate and Atmospheric Science, 3, 10.1038/s41612-020-00145-8, 2020.

Liu, D., Allan, J. D., Young, D. E., Coe, H., Beddows, D., Fleming, Z. L., Flynn, M. J., Gallagher, M. W., Harrison, R. M., Lee, J., Prevot, A. S. H., Taylor, J. W., Yin, J., Williams, P. I., and Zotter, P.: Size distribution, mixing state and source apportionment of black carbon aerosol in London during wintertime, Atmospheric Chemistry and Physics, 14, 10061-10084, 10.5194/acp-14-10061-2014, 2014.

Liu, D., Joshi, R., Wang, J., Yu, C., Allan, J. D., Coe, H., Flynn, M. J., Xie, C., Lee, J., Squires, F., Kotthaus, S., Grimmond, S., Ge, X., Sun, Y., and Fu, P.: Contrasting physical properties of black carbon in urban Beijing between winter and summer, Atmospheric Chemistry and Physics, 19, 6749-6769, 10.5194/acp-19-6749-2019, 2019.

Petit, J. E., Favez, O., Albinet, A., and Canonaco, F.: A user-friendly tool for comprehensive evaluation of the geographical origins of atmospheric pollution: Wind and trajectory analyses, Environmental Modelling & Software, 88, 183-187, 10.1016/j.envsoft.2016.11.022, 2017.

Syakur, M. A., Khotimah, B. K., Rochman, E. M. S., and Satoto, B. D.: Integration K-Means Clustering Method and Elbow Method For Identification of The Best Customer Profile Cluster, IOP Conference Series: Materials Science and Engineering, 336, 012017, 10.1088/1757-899X/336/1/012017, 2018.